# Predatory aggression evolved through adaptations to noradrenergic circuits

Güniz Göze Eren[1,5], Leonard Böger[1,2,5], Marianne Roca[1], Fumie Hiramatsu[1], Jun Liu[2], Luis Alvarez[2], Desiree L. Goetting[1], Lewis A. Cockram[1], Nurit Zorn[1], Ziduan Han[3], Misako Okumura[4], Monika Scholz[2✉] & James W. Lightfoot[1✉]

Behaviours are adaptive traits evolving through natural selection. Crucially, the genetic, molecular and neural modifications that shape behavioural innovations are poorly understood[1]. Here, we identify specialized adaptations linked to the evolution of invertebrate aggression[2]. Using the predatory nematode *Pristionchus pacificus*, we developed a machine learning model from behavioural tracking data and identified robust behavioural states associated with aggressive episodes. Strikingly, predatory aggression coincides with a rewiring of key circuits across nematode evolution. We find modifications to the noradrenergic pathway, with octopamine promoting aggressive predatory bouts whereas tyramine antagonistically induces passive states. Modulation occurs through the octopamine receptors *Ppa-ser-3* and *Ppa-ser-6*, and tyramine receptor *Ppa-lgc-55*. These localize to sensory neurons whose inhibition diminishes aggressive events. Crucially, this octopaminergic innovation emerged within this predatory lineage, consistent with an ancient divergence in function. Thus, evolutionary adaptations in noradrenergic circuits facilitated the emergence of aggressive behavioural states associated with complex predatory traits.

Natural selection favours behavioural traits that enhance an organism's fitness. This process results in a wide range of behaviours adapted to specific ecological niches and facilitates the evolution of behavioural diversity across species[1]. Certain behaviours such as foraging, mating, predator avoidance and aggression are subject to strong environmental pressures and are therefore particularly susceptible to evolutionary change. Crucially, our nascent understanding of these complex processes stems from interspecies comparative studies. These include comparisons of *Peromyscus* mice which have evolved distinct burrowing behaviours for predator avoidance and parental care[3,4], as well as behavioural differences between diverse *Drosophila* species which influence courtship signals and foraging preferences[5–8]. Despite this progress, the precise genetic, molecular and neural bases of behavioural adaptations are poorly understood. Nematodes with their small nervous systems and well-developed genetic and molecular tools are potent systems for understanding evolutionary behavioural processes in detail. Compared with *Caenorhabditis elegans*, the predatory nematode *Pristionchus pacificus* has evolved striking behavioural differences. This includes diversification in odorant sensitivity[9], social preference[10,11] and aggressive behaviours used to establish territory and remove competitors from their environment[12–15]. Consequently, we have leveraged the behavioural diversity between these nematode species to investigate the evolutionary adaptations moulding *P. pacificus* aggressive traits.

*P. pacificus* is an omnivorous nematode species which, in addition to feeding on bacteria, also preys on the larvae of other nematodes (Fig. 1a and Supplementary Video 1). Importantly, although predation can be used by *P. pacificus* to generate an extra nutrient source, these interactions also show hallmarks of aggression. More specifically, alongside the killing of other species and the cannibalism of conspecifics for nutrients[14–16], killing is often not followed by consumption and instead can be used as an aggressive behaviour to remove rivals[13]. Furthermore, aggressive biting without killing also serves to displace competitors from a shared food source[12]. These behaviours are accomplished through the presence of teeth-like armaments and distinct pharyngeal dynamics that facilitate the puncturing of the prey cuticle and feeding on their innards[16–18]. However, it is evident that not every contact with potential prey results in an attack, indicating that *P. pacificus* predatory aggression is not indiscriminate (Supplementary Videos 1 and 2). Therefore, to investigate the mechanism underlying the evolution of these behaviours and their regulation in *P. pacificus*, we developed a high-throughput analysis tool to automatically characterize *P. pacificus* behaviour.

## Behavioural tracking and state prediction

Nematodes feed through their pharynx, a neuromuscular organ in the animal head. In *C. elegans*, a fluorophore label targeted to the pharyngeal muscle has facilitated the dissection of its feeding and locomotion behaviours across development and under different environmental conditions[19]. To track these behaviours in *P. pacificus*, we exploited a similar pharyngeal fluorophore label using the *Ppa-myo-2* promoter (Fig. 1b). This plasmid was integrated into the *P. pacificus* genome with no adverse behavioural effects (Extended Data Fig. 1). We observed much lower levels of *Ppa-myo-2* expression in the terminal bulb of the

[1]Max Planck Research Group Genetics of Behavior, Max Planck Institute for Neurobiology of Behavior – caesar, Bonn, Germany. [2]Max Planck Research Group Neural Information Flow, Max Planck Institute for Neurobiology of Behavior – caesar, Bonn, Germany. [3]State Key Laboratory for Crop Stress Resistance and High-Efficiency Production, College of Plant Protection, Northwest A&F University, Yangling, China. [4]Graduate School of Integrated Sciences for Life, Hiroshima University, Higashi-Hiroshima, Japan. [5]These authors contributed equally: Güniz Göze Eren, Leonard Böger. ✉e-mail: monika.scholz@mpinb.mpg.de; james.lightfoot@mpinb.mpg.de

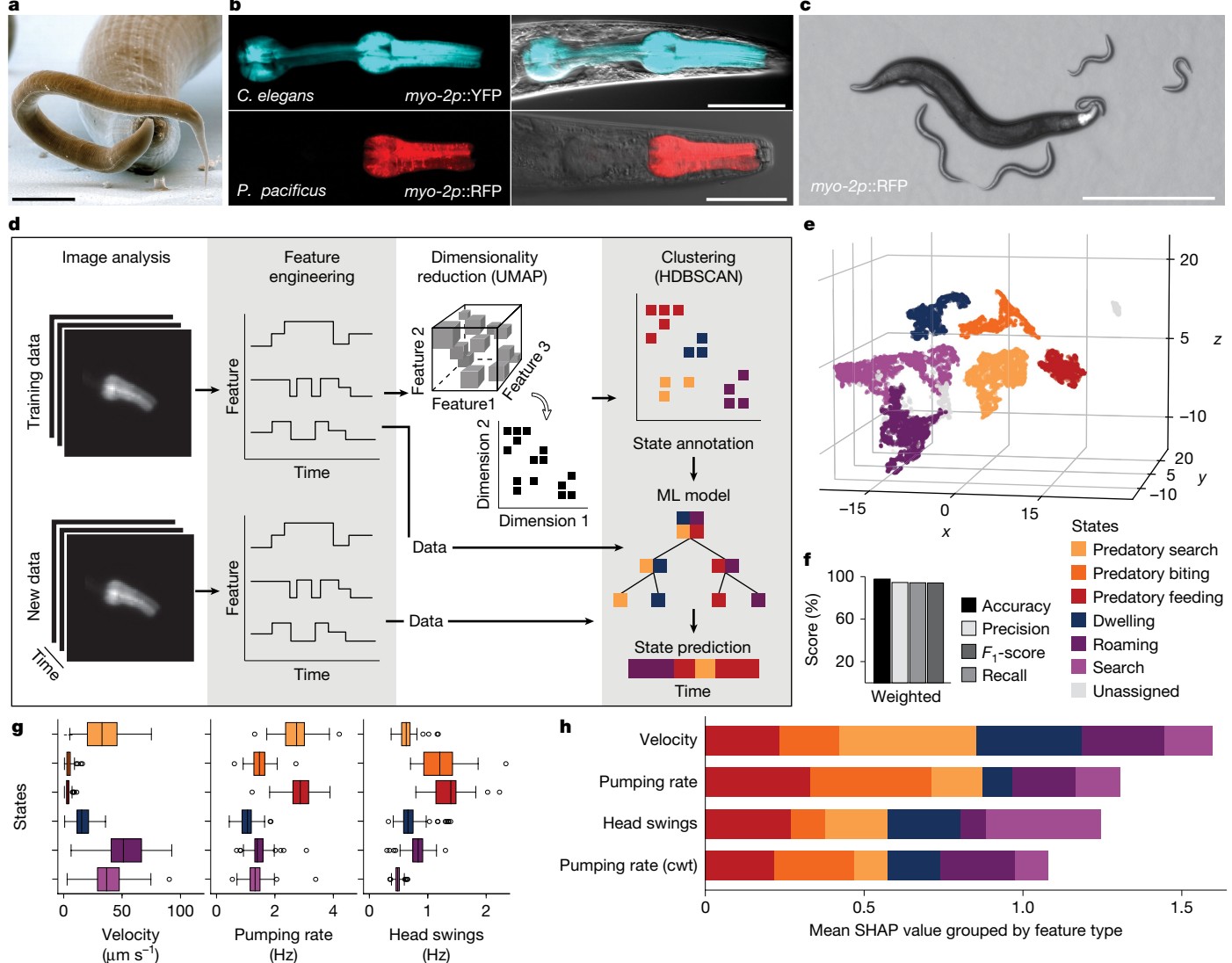

**Fig. 1 | Machine learning model predicts behavioural states from high-throughput tracking data of a predatory nematode. a**, Scanning electron microscopy image of *P. pacificus* (background nematode) with *C. elegans* larvae (foreground nematode). **b**, *myo-2p*::YFP expression in *C. elegans* compared with *myo-2p*::RFP expression in *P. pacificus*. **c**, Predatory *P. pacificus* animal surrounded by larval *C. elegans* prey. **d**, Schematic of the machine learning pipeline used to classify behavioural states. ML, machine learning; UMAP, uniform manifold approximation and projection. **e**, UMAP embedding of behavioural features. Colours indicate the six behavioural states identified by hierarchical clustering. **f**, Performance metrics of the behavioural state classifier on new, unseen data using weighted metrics. **g**, Distribution of key behavioural features in each state. Each point in the box plots corresponds to the mean value per state and per tracked animal. Box plots follow Tukey's rule with the box from first to third quartiles, and a line at the median. The whiskers denote 1.5× interquartile range. **h**, Feature importance (Shapley additive explanations (SHAP) values) for each of the basic behavioural features involved in the model. Features derived from the same base feature are averaged. Colours correspond to the behavioural states represented in **e**. cwt, continuous wavelet transform. See the Methods for statistics. Scale bars, 20 μm (**a**), 50 μm (**b**), 500 μm (**c**).

*P. pacificus* pharynx than in *C. elegans* because of the enlarged gland cells found in this space along with the absence of a hardened grinder structure in Diplogastridae[20]. Despite this difference, we were able to use this method to successfully track feeding and locomotion in many animals simultaneously on standard assay conditions (Fig. 1c). These consist of *Ppa-myo-2p*::RFP-expressing *P. pacificus* predators placed onto an assay arena that contains either *C. elegans* larvae in abundance as potential prey or a bacterial lawn as a food source. During behavioural tracking, we extracted multiple features including speed, reversals and feeding events, as well as posture-related measures using the image analysis tool PharaGlow[19]. To automatically identify predation-associated states, we employed a machine learning pipeline combining low-dimensional embedding and hierarchical clustering (Fig. 1d). Similar pipelines have been used for unsupervised state classification of behavioural tracking data in flies, mice, fish and *C. elegans*[21–27]. Using

this approach, we found six distinct behavioural states (Fig. 1e and Extended Data Fig. 2a,b). We subsequently identified the behaviours in each state on the basis of the overlap with annotations of an expert human annotator (Extended Data Fig. 2c). Several of these seem to correspond to canonical states also observed in *C. elegans*, including 'roaming' and 'dwelling' states; however, we also identified three new behavioural states that correlate to a predatory environment. These states are labelled 'predatory search', 'predatory biting' and 'predatory feeding'. To further validate these states, we applied the same pipeline to an independent dataset, using the same set of parameters for dimensionality reduction. We again found the same six behavioural states as in the original dataset, indicating these clusters are consistent and robust across independent experiments (Extended Data Fig. 2d–f). Next, to extend the model to unseen data, we fit an XGBoost multi-class classifier[28] to the clustered data, and analysed its performance on a test set of

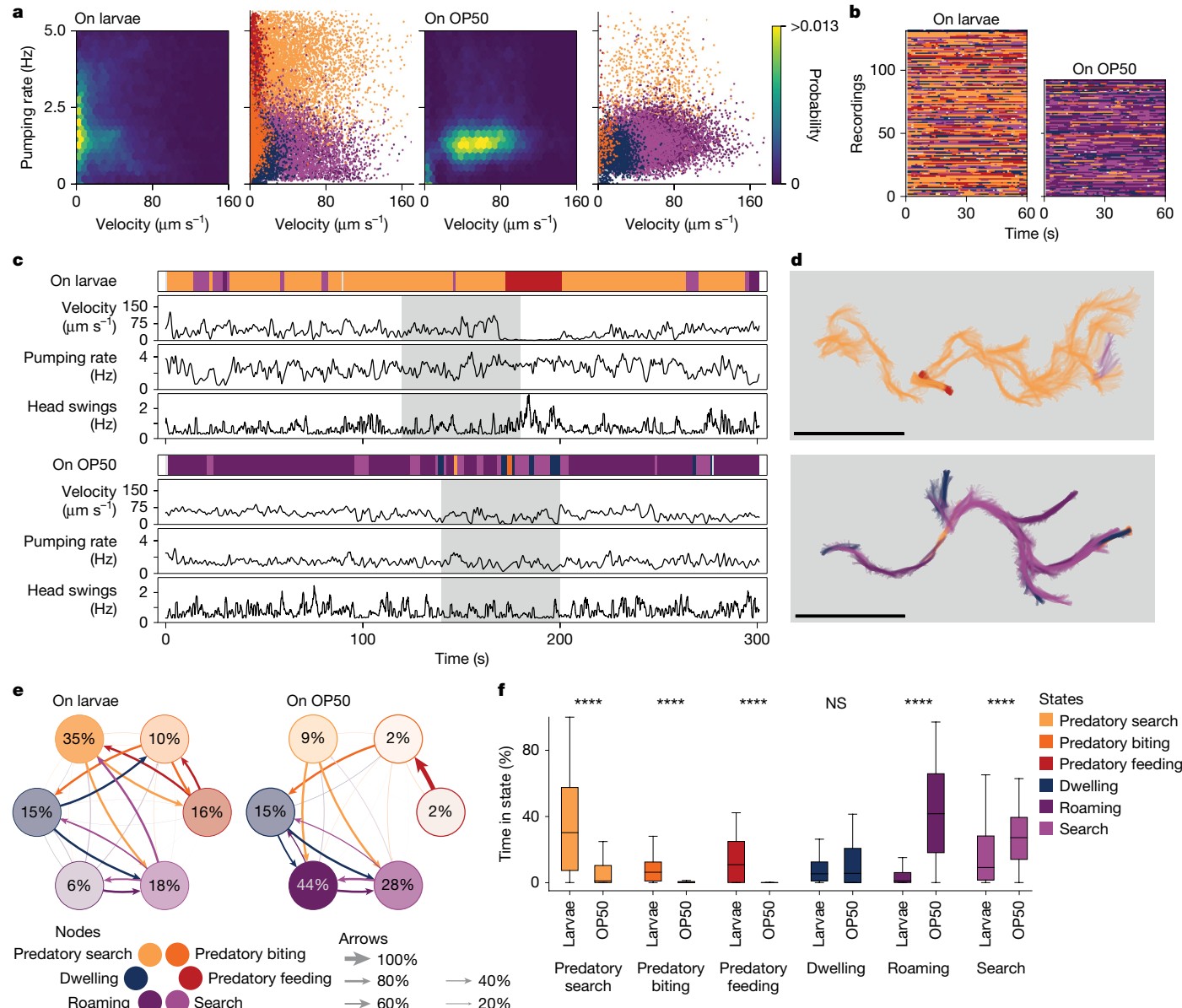

**Fig. 2 | Automatic classification of behavioural data reveals context-dependent predation drive. a**, Probability density map of velocity and pumping rate for animals on larval prey or OP50 bacteria. Scatter plots indicate the corresponding state assignments. **b**, Ethograms showing predicted behavioural states for animals on prey (left) or bacteria (right). **c**, Ethograms, velocity, pumping rate and head swings for a representative animal on larval prey (top) or bacterial food (bottom). **d**, Pharyngeal centreline as coloured by the assigned behavioural state. The tracks shown correspond to the grey regions in **c**. **e**, Average transition rates between behavioural states for animals on larval prey and bacterial food, respectively. Numbers in circles indicate the

fraction of time per state. Arrow thickness indicates the transition rate normalized to outgoing transitions. **f**, Mean fraction of time spent in each behavioural state per animal. Box plots follow Tukey's rule with the box from first to third quartiles, and a line at the median. The whiskers denote 1.5× interquartile range. Statistics were performed with Mann–Whitney $U$-test using Bonferroni correction for multiple tests against WT on larvae. ****$P < 0.0001$; $n = 131$ for WT on larvae, $n = 92$ for WT on OP50; NS, not significant. Minimum of three biological replicates. Statistics, sample size and $P$ values are available in Supplementary Table 3. Scale bars, 250 μm (**d**).

held-out recordings. The model captures the cluster labels with greater than 95% in both accuracy and recall (Fig. 1f and Extended Data Fig. 3), with velocity, pumping rate and head swings contributing most to state identity (Fig. 1g,h). Consequently, this pipeline allowed us to predict the behavioural states for unrestrained animals (Supplementary Video 3), and provides a mechanism to investigate the molecular determinants driving the evolution of predatory aggression in *P. pacificus*.

## Context modulates predatory drive

To determine the specificity of our identified behavioural states, we investigated the influence of the environmental and sensory context

on *P. pacificus* state occupancy by analysing approximately 10.5 animal hours of tracking data on either a bacterial food source or larval prey. As the most descriptive features for each state are velocity and pumping rate of the animals (Fig. 1h and Extended Data Fig. 3d), the joint distribution of these two features showed distinct clusters when animals were placed on prey larvae or on bacterial food (Fig. 2a,b). By comparing these density plots with the identified behavioural states, we observed that animals preferentially occupy predation-related states when placed on larvae (Fig. 2b), whereas they spend more time in low-pumping-rate states with higher speeds when exposed to bacteria alone (Fig. 2b, right). We identified two types of search states, which we label as 'search' and 'predatory search'. Both are characterized by

higher speeds; however, on prey, they showed exaggerated head swing amplitudes during 'predatory search' making this state distinct from the 'search' state, which is more prominent on bacterial foods (Fig. 2c,d). These findings suggest that sensory context dictates state occupancy such that predatory states are nearly unique to prey-rich environments, further validating our previous annotations (Fig. 2c–f and Extended Data Fig. 2c). Interestingly, we found that although transitions between states and the total time spent in each state are dependent on sensory contexts, the average duration of a behavioural state remained similar for most states (Fig. 2e,f and Extended Data Fig. 4a–c). We therefore focused on the total time spent in each state as a measure of the propensity for predatory and aggressive behaviour versus non-predatory behaviour.

Given that predation relies on nose contact with prey, we provided direct evidence for predatory-associated states by observing predator–prey interactions using a dual-colour tracking epifluorescence microscope[29]. We tracked individual *P. pacificus* predators placed in an environment containing *C. elegans* prey that express GFP in their body wall muscles (Extended Data Fig. 5a). Using this method, we were able to predict behavioural states of the predator from its behavioural features (Extended Data Fig. 5b,c and Supplementary Video 4) and simultaneously observe the location of the prey without considering prey information in the model. By collating and aligning all predicted biting events from multiple datasets, we found that the prey signal rose shortly before the onset of biting, confirming that animals were in contact with prey during 'predatory biting' states (Extended Data Fig. 5d,e). Furthermore, as parts of the prey animal were labelled, we could also observe ingestion by following the fluorescent signal of food that moved from the mouth towards the intestine (Extended Data Fig. 5f and Supplementary Video 5). These events correlated with the predicted 'predatory feeding' state, reinforcing our model prediction.

As predatory biting fundamentally serves two functions, nutrient-driven behaviour and aggression, we wanted to further dissect the impact of aggression on the 'predatory biting' state which we use as a proxy for predatory aggression throughout. Nutrition-driven biting should be followed by 'predatory feeding', which is ultimately the state in which material is ingested (Extended Data Fig. 5f). By contrast, when 'predatory biting' occurs without consecutive feeding, we expect these bites to have primarily an aggressive function. To differentiate distinct function in our data, we compared behavioural state changes in *P. pacificus* animals surrounded by larvae with those surrounded by both larvae and a bacterial food source.

Here, although we found that the time spent in the 'predatory biting' state remained relatively consistent between these conditions, animals exposed to both bacteria and larvae spent less time in the 'predatory feeding' state (Extended Data Fig. 5g). In addition, transitions from 'predatory biting' to 'predatory feeding' compared with larvae alone were also reduced (Extended Data Fig. 5h,i). Taken together, these data confirm that a greater proportion of contacts are aggressive in nature and non-nutritionally associated when both food choices are available, supporting the interpretation that predatory biting also serves an aggressive drive. Thus, we observe a greatly expanded state complexity in *P. pacificus*, going beyond the canonical foraging switch between roaming and dwelling found in *C. elegans*. This complexity may be a feature of the *P. pacificus* omnivorous lifestyle and dietary switching, or instead a characteristic of the evolution of predation.

## Noradrenergic modulation of aggression

Persistent behavioural states are often stabilized by distinct neuromodulators. Frequently, these act antagonistically to set mutually exclusive patterns of behaviour, for example, by regulating the duration of opposing states such as sleep and wakefulness, or hunger and satiety[30–33]. Therefore, to explore whether similar circuits are involved in establishing and maintaining the aggressive predatory states versus the

non-predatory states, we screened through mutants of the four major neuromodulators using strains generated by means of CRISPR–Cas9 (Supplementary Table 1)[34,35]. These have one-to-one orthology with their *C. elegans* counterparts (Extended Data Fig. 6a). Mutations generated in *Ppa-tph-1*, *Ppa-cat-2* and *Ppa-tbh-1* affect serotonin, dopamine and octopamine production, respectively, whereas *Ppa-tdc-1* affects the production of both tyramine and octopamine, as these neuromodulators are in the same biochemical synthesis pathway (Fig. 3a). All mutants were crossed into the *Ppa-myo-2p*::RFP background and assessed using the high-throughput tracking and machine learning pipeline previously established.

In *C. elegans*, serotonin plays a multifaceted role in its behaviour including regulating feeding, locomotion, foraging, egg laying, stress response, and learning and memory[36–38]. In *P. pacificus*, previous work identified a further predation-specific role for this neurotransmitter in synchronizing the action of the tooth and pharyngeal pumping, proving essential for efficient predation[35]. Correspondingly, *Ppa-tph-1* mutants show increased roaming behaviours similar to observations in *C. elegans* and a strong decrease in predation consistent with previous findings (Fig. 3b–d and Extended Data Fig. 6b). Similar to *Ppa-tph-1*, the *Ppa-cat-2* dopamine-synthesis-deficient animals also show a change in body posture and movement speed (Fig. 3b,c and Extended Data Fig. 6b); however, they do not exhibit the decrease in 'predatory biting' observed in *Ppa-tph-1* serotonin mutants. Instead, we see a reduction in the 'predatory search' and 'predatory feeding' states (Fig. 3d). In *C. elegans* dopamine is critical for efficient foraging when food is present[36,39], and we predict that in *P. pacificus*, dopamine may be required for initiating feeding behaviour post successful kill which may relate to a food reward signal.

Although the effects of serotonin and dopamine are well described in *C. elegans*, much less is known regarding the function of the noradrenergic neuromodulators tyramine and octopamine. Tyramine has been implicated in modulating a rapid 'flight' escape response by linking head movements with locomotion and also plays a role in other long-term stress responses[40–43]. By contrast, octopamine initiates a fasting signal facilitating exploration and optimizing foraging strategies during nutrient scarcity[44–46]. In *P. pacificus*, we detect significantly reduced levels of the 'predatory biting' state in *Ppa-tbh-1* mutants and fewer transitions into this state, indicating octopamine promotes predatory aggression in *P. pacificus* (Fig. 3b–e). Strikingly, although *Ppa-tdc-1* is required for the biosynthesis of both tyramine and octopamine, in *Ppa-tdc-1* mutants predatory-associated states are maintained at wild-type (WT) levels. This indicates the further loss of tyramine suppresses the octopamine-induced predation defect (Fig. 3b–e and Extended Data Fig. 6c,d). A similar finding is observed in *Ppa-tdc-1; Ppa-tbh-1* double mutants which phenocopy the *Ppa-tdc-1* mutants (Extended Data Fig. 6e–h). In line with these data, applying exogenous tyramine to WT animals also induces non-predatory bouts whereas the addition of exogenous octopamine maintains high levels of predation in WT animals and rescues the low predatory aggression phenotype observed in *Ppa-tbh-1* mutants (Extended Data Fig. 7a–f). Thus, in *P. pacificus*, these neuromodulators regulate a new behaviour absent in *C. elegans*. Octopamine enables robust and prolonged predatory behaviours associated with an aggressive drive whereas tyramine acts antagonistically to establish the docile, non-predatory state.

## Evolution rewires noradrenergic networks

Having established the functional differences associated with these neuromodulators in *P. pacificus*, we next explored the neural circuits associated with their biosynthesis. In *C. elegans*, the interneurons RIM and RIC represent the only tyraminergic neurons, and they express *Cel-tdc-1*, although the RIC neurons additionally express *Cel-tbh-1* and are therefore also octopaminergic[41]. To investigate whether the expression of these enzymes and the potential circuits involved in *P. pacificus* are

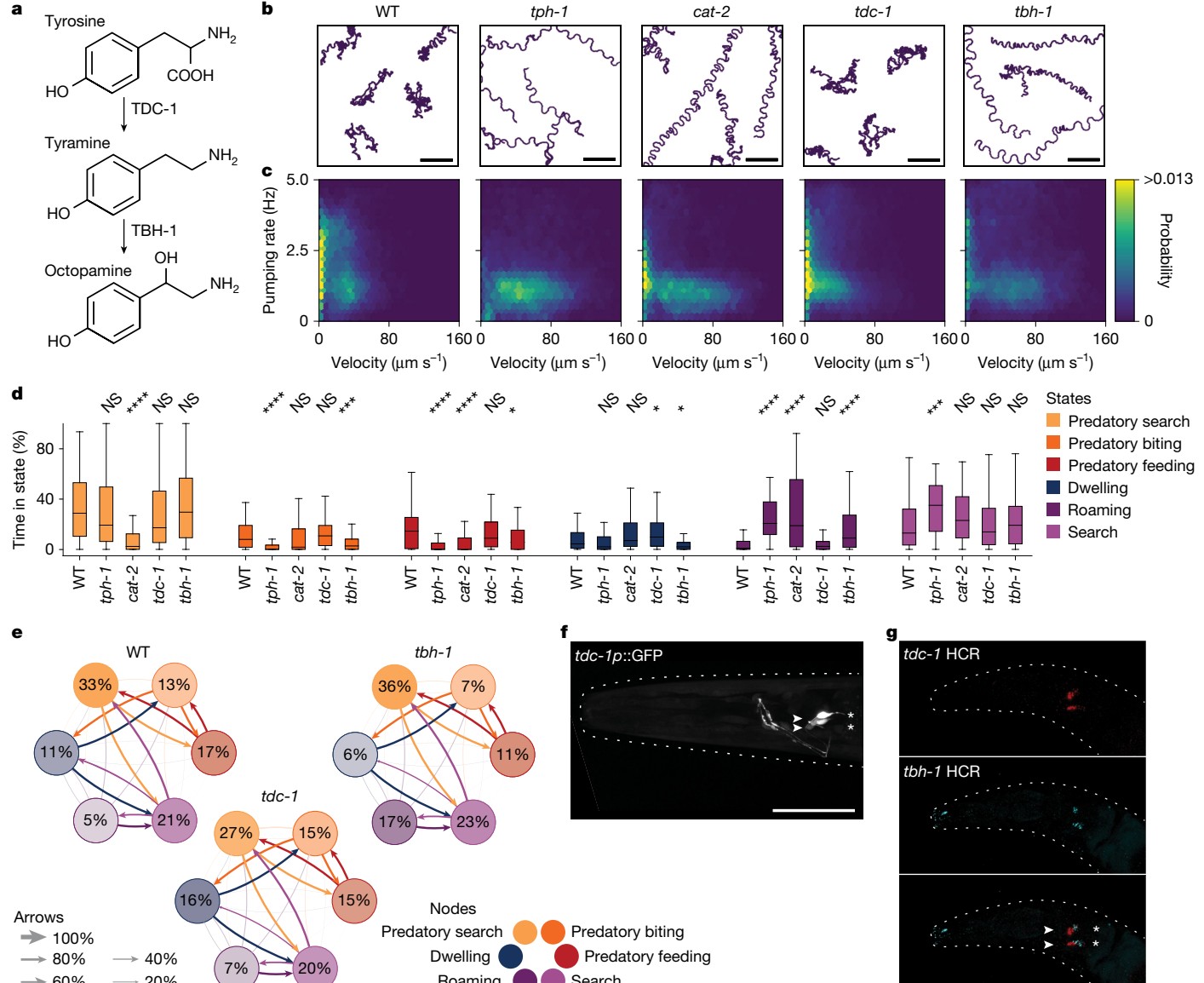

**Fig. 3 | Noradrenergic systems modulate predatory aggression. a**, Synthesis pathway of tyramine and octopamine from the precursor tyrosine. The enzymes involved in tyramine (TDC-1) and octopamine (TBH-1) synthesis act in the same pathway. **b**, Example animal tracks for WT and *tph-1*, *cat-2*, *tdc-1* and *tbh-1* mutants. **c**, Probability density map of velocity and pumping rate for animals corresponding to the genotypes in **b**. **d**, Time spent in each behavioural state normalized to the total track duration. Box plots follow Tukey's rule with the box from first to third quartiles, and a line at the median. The whiskers denote 1.5× interquartile range. Statistics were performed with Mann–Whitney *U*-test using Bonferroni correction for multiple tests against WT. *$P < 0.05$, ***$P < 0.001$, ****$P < 0.0001$; $n = 91$ for WT, $n = 56$ for *Ppa-tph-1*, $n = 97$ for *Ppa-cat-2*, $n = 167$ for

*Ppa-tdc-1*, $n = 141$ for *Ppa-tbh-1*. Minimum of three biological replicates. **e**, Average transition rates between behavioural states for WT and *tdc-1* and *tbh-1* mutants on larval prey. The number in circles indicates the average state duration as in **d** and the arrow size indicates the transition rate normalized to outgoing transitions. **f**, *tdc-1p*::GFP expression in *P. pacificus*. Arrows indicate a putative pair of RIM neurons whereas *indicates a putative pair of RIC neurons. **g**, Messenger RNA of *tdc-1* (red) and *tbh-1* (cyan) and colocalization visualized using HCR. Arrows indicate a putative pair of RIM neurons whereas *indicates a putative pair of RIC neurons. See the Methods for statistics. Scale bars, 1 mm (**b**), 50 μm (**f**), 25 μm (**g**).

conserved, we generated a transgenic strain expressing *tdc-1p*::GFP (Fig. 3f). Similar to *C. elegans*, we detected two pairs of neurons that we putatively identified as the *P. pacificus* RIM (anterior) and RIC (posterior) neurons. To clarify the identity of these neurons further, we also compared the soma shape, position and neurite projections of these cells with the known morphology of the *P. pacificus* RIM and RIC neurons acquired from the recently published head connectome[47]. All of these features aligned closely with the neurons in our dataset. In addition, we used in situ hybridization chain reaction (HCR) to investigate the presence of *tdc-1* and *tbh-1* transcripts in *P. pacificus*. As in *C. elegans*, we detected *Ppa-tdc-1* transcripts in two pairs of neurons that owing

to soma position and neurotransmitter expression we identified as the *P. pacificus* equivalent of RIM and RIC. Additionally, we detected *Ppa-tbh-1* transcripts in a single neuron pair that co-localized with the posterior *Ppa-tdc-1*-positive neuron that we identified as RIC (Fig. 3g). Importantly, our identification of these cells is consistent with recent findings describing many of the monoaminergic neurons in *P. pacificus*[48]. Subsequently, we also attempted to confirm that these cells are the relevant neurons producing the bioamines involved in predatory aggression by genetically silencing them. In *P. pacificus*, expression of a histamine-gated chloride channel (HisCl) enables the inducible inhibition of targeted neurons upon the addition of

exogenous histamine. Accordingly, we used the *tdc-1* promoter to drive expression of HisCl in both RIM and RIC and attempted to recapitulate the *tdc-1* mutant phenotype. Silencing of both neurons resulted in a partial rescue of the reduced predatory aggression phenotype observed in *tbh-1* mutants. This is similar to the *tdc-1* mutant rescue, confirming these cells are the probable functional origin of octopamine and tyramine (Extended Data Fig. 8). Thus, the production of tyramine and octopamine in the RIM and RIC neurons is probably conserved between *C. elegans* and *P. pacificus* although the neuromodulatory function has diverged.

Next, we attempted to elucidate the receptor circuits to further understand the molecular mechanisms involved in generating the aggressive predatory and non-predatory states. In *C. elegans*, three octopamine receptors have been identified, *Cel-ser-3*, *Cel-ser-6* and *Cel-octr-1* (refs. 49–51). We identified 1:1 orthologues in *P. pacificus* for all three receptors and generated CRISPR–Cas9 mutants in these genes in the *Ppa-myo-2p*::RFP strain (Extended Data Figs. 9 and 10a,b and Supplementary Tables 1 and 2). We then assessed the predatory aggressive drives of these animals using our behavioural state model pipeline. Although mutations in *Ppa-octr-1* maintained WT predatory biting, mutations in both *Ppa-ser-3* and *Ppa-ser-6* phenocopied the 'reduced predatory biting' state observed in *Ppa-tbh-1* mutant (Fig. 4a–c and Extended Data Fig. 10c). Therefore, both *Ppa-ser-3* and *Ppa-ser-6* are required to establish efficient *P. pacificus* predatory aggressive bouts through octopamine signalling. Similarly, there are four known tyramine receptors in *C. elegans*, *Cel-tyra-2*, *Cel-tyra-3*, *Cel-ser-2* and *Cel-lgc-55* (refs. 50,52–54). To identify potential tyramine receptors involved in this pathway, we also identified 1:1 orthologues in *P. pacificus* for all four of these receptors and generated corresponding CRISPR–Cas9 mutants in the *Ppa-tbh-1*; *Ppa-myo-2p*::RFP strain to determine whether any rescued the *Ppa-tbh-1* reduced-killing phenotype (Extended Data Figs. 9 and 10b and Supplementary Tables 1 and 2). Mutations in *Ppa-tyra-2*, *Ppa-tyra-3* and *Ppa-ser-2* as well as the corresponding triple mutant maintained low levels of predatory aggression similar to the *Ppa-tbh-1*. However, in *Ppa-lgc-55* mutants, predation was restored to similar levels as seen in WT and *Ppa-tdc-1* mutants (Fig. 4d–f and Extended Data Fig. 10d–g). Additionally, we observe a substantial increase in the 'predatory search' behaviour in *Ppa-tyra-3* mutants indicating a potential role for *Ppa-tyra-3* in regulating this behavioural state (Fig. 4d–f and Extended Data Fig. 10d–g). Thus, two octopamine receptors and a single tyramine receptor are required to mediate the predatory states and associated aggressive transitions.

To identify the neurons governing behavioural state-switching, we first used HCR to detect the transcripts of the *Ppa-ser-3*, *Ppa-ser-6* and *Ppa-lgc-55* receptors associated with the predatory states. Many head cells were positive for these transcripts making it difficult to determine cellular identification from transcript position alone (Extended Data Fig. 10h). Consequently, we also generated transcriptional reporter lines for all three receptors and compared these with reporter lines of the same receptors in *C. elegans*. The octopamine receptor *Ppa-ser-3* is expressed in the neck muscles, several head neurons and strikingly the IL2 and IL1 head sensory neurons with their distinctive neurites projecting to the worm's nose[47]. *Ppa-ser-6* is expressed in a non-overlapping set of head neurons which also includes a neuron pair with anterior processes (Fig. 4g). For the tyraminergic receptor *Ppa-lgc-55*, we again observe robust expression in the neck muscles which is similar to observations in *C. elegans*[52], but we also detect strong expression in a distinct set of head sensory neurons separate from the octopamine receptor expressing IL2s and IL1s (Fig. 4g). These are putatively identified as the OL neurons on the basis of neurite morphology and soma placement matching data from electron microscopy[47]. Using the CeNGEN dataset[55] and transcriptional reporter lines, we found these three receptors in *C. elegans* are expressed throughout a large subset of head neurons including sensory, inter and motor neurons (Extended Data Fig. 10i). However, there is minimal overlap with any of the head sensory neurons

we observe in *P. pacificus* (Fig. 4g). Therefore, in *P. pacificus* the octopamine and tyramine receptors regulating aggressive behavioural states are localized to head sensory neurons which are distinct from their localization in *C. elegans*.

## Silencing IL2 neurons disrupts predation

Some of the most distinctive neurons receiving octopamine signals in *P. pacificus* are the *Ppa-ser-3*-expressing IL2 sensory neurons. In *C. elegans*, these neurons project sensory endings into the environment and are associated with nictation behaviour and sensory modulation[56] but they do not express any octopamine receptors[55]. In *P. pacificus*, these are similarly environmentally exposed and are the first point of contact between predator and prey (Fig. 5a). Therefore, to investigate their role further, we silenced these neurons using an IL2-specific promoter. In *C. elegans*, *Cel-klp-6* is robustly expressed in only this group of six sensory neurons[55]. We confirmed that a *Ppa-klp-6* reporter also resulted in strong and specific IL2 expression in *P. pacificus* (Fig. 5b). Subsequently, we used this promoter to drive expression of HisCl to silence these neurons. Upon IL2 silencing, we observed a substantial decrease in the detection of predatory events and correspondingly the time in all predation-associated states was significantly reduced (Fig. 5c,d). Furthermore, transitions into predatory aggressive states were also less frequent (Fig. 5e and Supplementary Table 3). Thus, the IL2 neurons are essential for establishing general predatory aggression levels in *P. pacificus*.

## Lineage-specific evolution of aggression

Finally, although *P. pacificus* is the most well studied of the Diplogastridae, nearly all described species in this taxon are capable of predatory aggression[16]. Therefore, as octopamine and its receptors are essential for the generation of these behaviours in *P. pacificus*, we explored the evolutionary origins of this association. *Allodiplogaster sudhausi* is a large, free-living nematode species and is a basal member of this family (Fig. 5f). Similar to *P. pacificus*, it has acquired teeth-like structures and is a highly predatory and cannibalistic species (Fig. 5g). Furthermore, it recently underwent a whole genome duplication event which probably contributed to its enlarged size[57]. To investigate a conserved role for octopamine in predatory aggression across the Diplogastridae family, we generated mutations in both copies of *Asu-tbh-1* (Extended Data Fig. 9). We then screened for predatory aggression by directly counting the number of *C. elegans* corpses after larvae were exposed to either WT or *Asu-tbh-1* mutant predators. We detected fewer *C. elegans* corpses on assays with *Asu-tbh-1* mutants consistent with lower levels of aggression and fewer predatory interactions (Fig. 5h). Thus, our findings are consistent with an ancient divergence in octopamine function across nematode evolution that coincides with the origin of predatory aggression in the Diplogastridae.

Taken together, we propose a model whereby octopamine and tyramine antagonistically regulate aggressive state-switching (Fig. 5i). This occurs through state-dependent gating of head sensory neurons which, when stimulated, can be used to detect the presence of a contact event with prey and result in an aggressive predatory attack. Furthermore, this evolved early in the Diplogastridae, enabling the aggressive predatory behaviours found throughout these species.

## Discussion

Reconstructing the evolutionary changes generating new behaviours is dependent on combining neurobiological and behavioural studies between phylogenetically well-characterized species. Accordingly, the divergent behaviours observed between *C. elegans* and *P. pacificus* are well suited to uncovering innovations associated with behavioural evolution. In our study, we focused on the evolution of *P. pacificus* aggressive

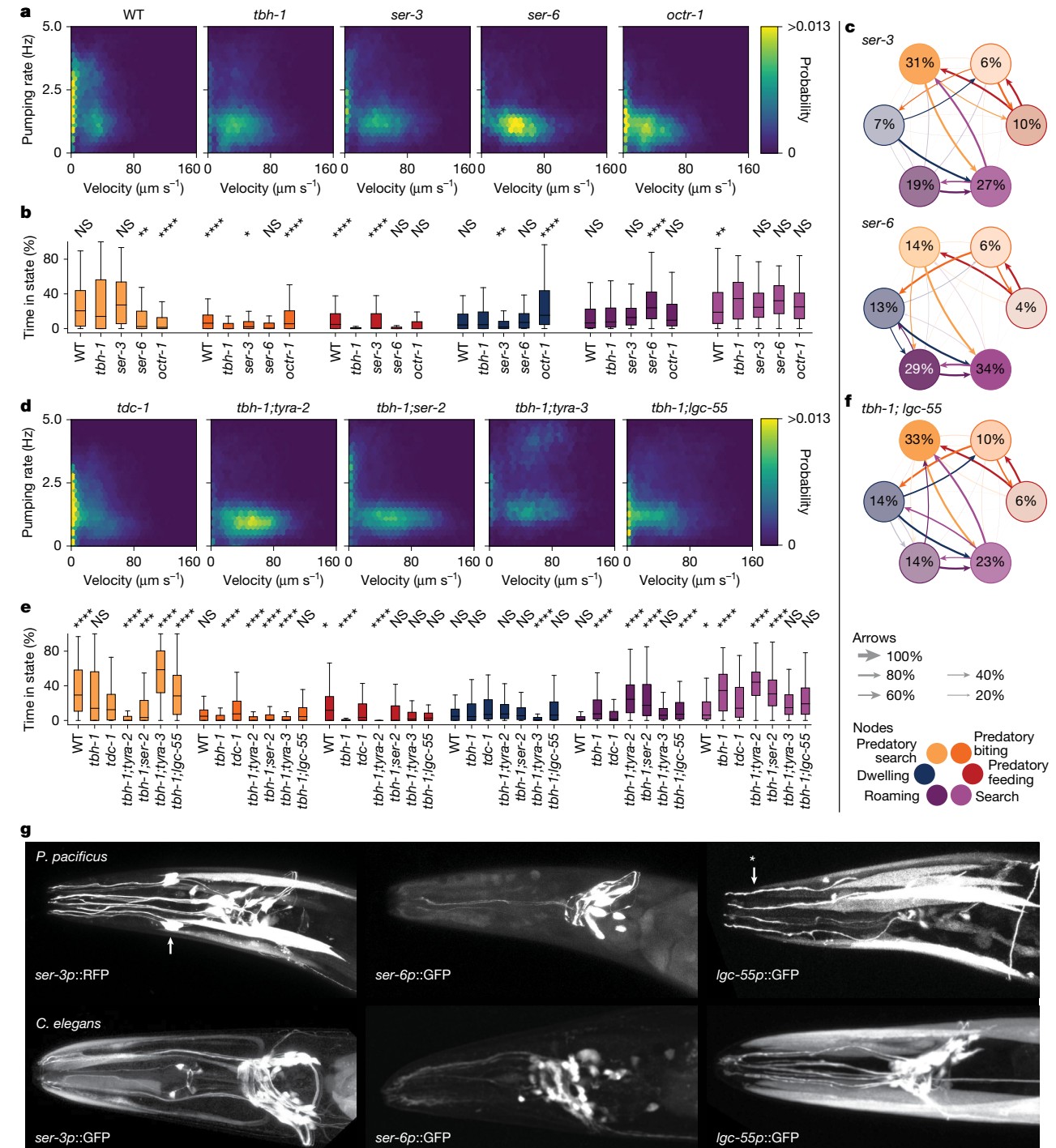

**Fig. 4 | Octopamine and tyramine receptors in sensory neurons gate aggressive state entry and exit. a**, Probability density map of velocity and pumping rate for WT, *tbh-1* and the octopamine receptors *ser-3*, *ser-6* and *octr-1*. *ser-3* and *ser-6* phenocopy *tbh-1*. **b**, Relative time in each behavioural state for all genotypes in **a**. Statistics were performed with Mann–Whitney *U*-test using Bonferroni correction for multiple tests against *Ppa-tbh-1*. *n* = 170 for WT, *n* = 197 for *Ppa-tbh-1*, *n* = 137 for *Ppa-ser-3*, *n* = 104 for *Ppa-ser-6*, *n* = 215 for *Ppa-octr-1*. Minimum of three biological replicates. **c**, Average transition rates between behavioural states for *ser-3* and *ser-6*. The number in circles indicates the average state duration as in **b**, and the arrow size indicates the transition rate normalized to outgoing transitions. **d**, Probability density map of velocity and pumping rate for *tdc-1* and the tyramine receptors *tyra-2*, *ser-2*, *tyra-3* and *lgc-55*, all in the *tbh-1* background. **e**, Relative time in each behavioural state for all genotypes in **d**. Box plots follow Tukey's rule with the box from first to third quartiles, and a line at the median. The whiskers denote 1.5 × interquartile range. Statistics were performed with Mann–Whitney *U*-test using Bonferroni correction for multiple tests against *Ppa-tdc-1*. *n* = 116 for WT, *n* = 197 for *Ppa-tbh-1*, *n* = 235 for *Ppa-tdc-1*, *n* = 184 for *Ppa-tbh-1-tyra-2*, *n* = 298 for *Ppa-tbh-1-ser-2*, *n* = 60 for *Ppa-tbh-1-tyra-3*, *n* = 175 for *Ppa-tbh-1-lgc-55*. Minimum of three biological replicates. **f**, Average transition rates between behavioural states for *tbh-1; lgc-55*. The number in circles indicates the average state duration as in **e** and the arrow size indicates the transition rate normalized to outgoing transitions. **g**, Comparative expression pattern analysis for the octopamine receptors *ser-3* and *ser-6*, as well as the tyramine receptor *lgc-55*, in *P. pacificus* (top) and *C. elegans* (bottom). Arrow indicates putative IL2 neurons in *P. pacificus*. Arrow with * indicates putative OL cell neurites in *P. pacificus*. Scale bar, 50 μm. See the Methods for statistics. *$P < 0.05$, **$P < 0.01$, ***$P < 0.001$, ****$P < 0.0001$.

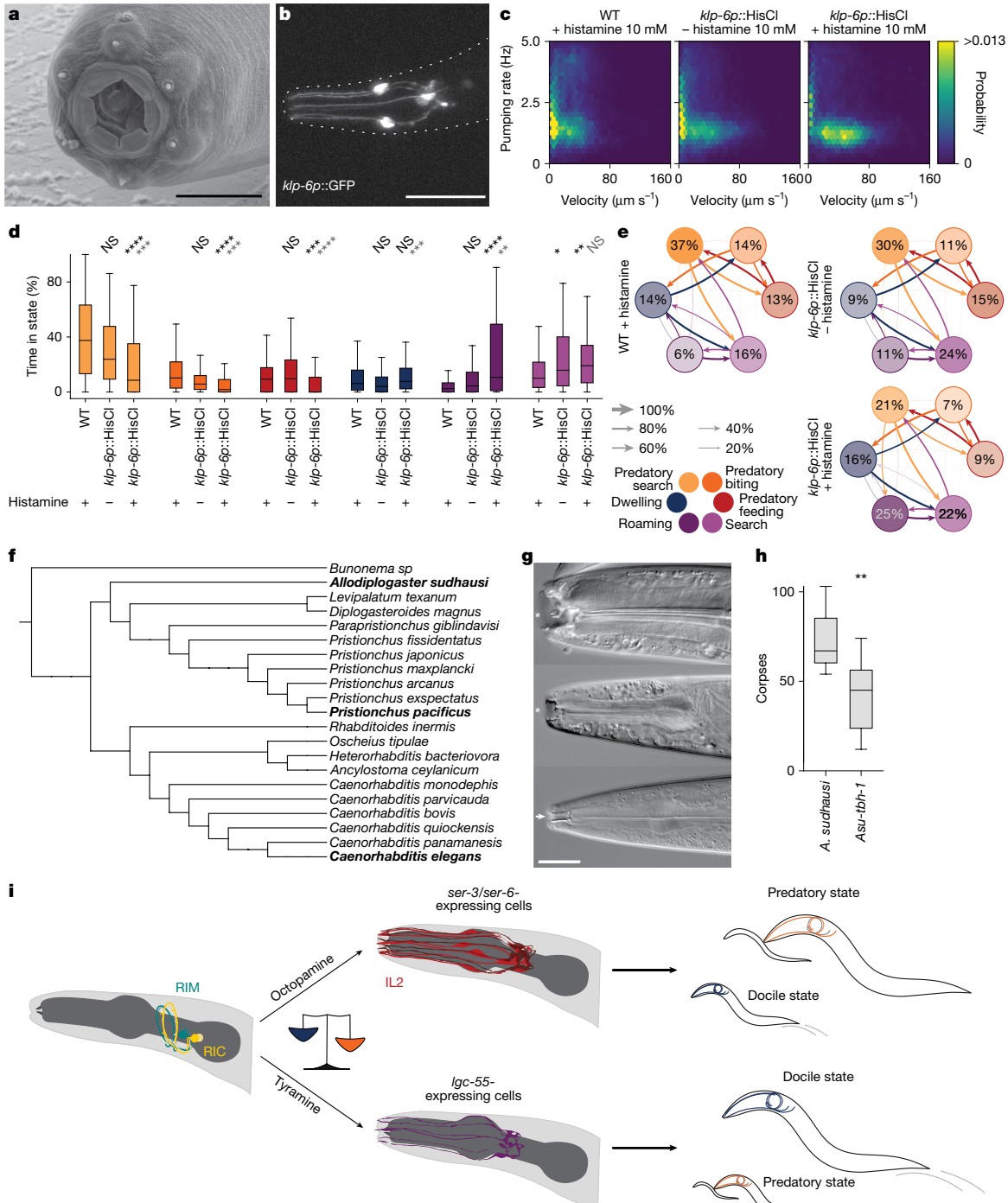

**Fig. 5 | Predatory aggression is dependent on conserved sensory neuron function. a**, Scanning electron microscopy image of the *P. pacificus* face. Six sensory endings from the IL2 neurons circle the mouth opening and are candidates for prey detection. **b**, *klp-6p*::GFP expression is specifically localized to IL2 neurons in *P. pacificus*. **c**, Probability density map of velocity and pumping rate for genetic silencing of IL2 neurons. Plots show WT + 10 mM histamine and *klp-6p*::HisCl without histamine controls and *klp-6p*::HisCl + 10 mM histamine silencing conditions. **d**, Relative time in each behavioural state for all conditions in **c**. Box plots follow Tukey's rule with the box from first to third quartiles, and a line at the median. The whiskers denote 1.5× interquartile range. Statistics were performed with Mann–Whitney *U*-test using Bonferroni correction for multiple tests, against WT (black asterisks) and against *Ppa-klp-6*::HisCl − histamine (grey asterisks). *n* = 103 for WT, *n* = 107 for *Ppa-klp-6*::HisCl − histamine, *n* = 104 for *Ppa-klp-6*::HisCl + histamine. Minimum of three biological replicates. **e**, Average transition rates between behavioural states for WT + 10 mM histamine,

*klp-6p*::HisCl without histamine and *klp-6p*::HisCl + 10 mM histamine silencing conditions. The number in circles indicates the average state duration as in **d**, and the arrow size indicates the transition rate normalized to outgoing transitions. **f**, Schematic phylogeny illustrating the evolutionary relationships among nematodes. Bold text indicates selected comparative species of interest. **g**, Differential interference contrast mouth images of the predatory *A. sudhausi* (top) and *P. pacificus* (middle) with mouths containing teeth-like structures (*) and the microbial feeder *C. elegans* (bottom) with an empty buccal cavity (arrow). **h**, Corpse assays of *A. sudhausi* WT compared with *Asu-tbh-1* mutants that show reduced predatory aggression. *C. elegans* corpses were counted after 2 h of exposure to five predators. Statistics were performed with Mann–Whitney *U*-test. *n* = 10 for *A. sudhausi*, *n* = 10 for *Asu-tbh-1*. **i**, Proposed model of the octopamine/tyramine regulation driving aggressive versus docile behavioural states. Scale bars, 5 μm (**a**), 50 μm (**b**), 25 μm (**g**). *$P < 0.05$, **$P < 0.01$, ***$P < 0.001$, ****$P < 0.0001$.

behaviours that have not been previously reported in other nematodes including *C. elegans*. In *P. pacificus*, aggression is a state associated with the regulation of multiple behaviours including predation, cannibalism and territoriality[12–15]. We found that this aggression is regulated through the actions of octopamine and tyramine, which share similarities with known attributes of aggression in several other invertebrate species. This includes territorial behaviours and nestmate recognition in social insects, intraspecific and intersexual cannibalism in arachnids, and fighting in crustaceans[2]. Similarly, in *D. melanogaster* aggression has been shown to depend on a complex system of neurotransmitters including octopamine as well as other hormones and neuropeptides[58,59].

In *C. elegans*, octopamine and tyramine have evolved independent functions, with tyramine broadly regulating the escape response and octopamine inducing fasting-associated behaviours[40–45]. By contrast, our study shows that during the evolution of *P. pacificus*, the function of these neuromodulators has become tightly interlinked and that they instead act antagonistically to regulate its predatory aggression. Additionally, we found that the distribution of octopamine and tyramine receptors has greatly diverged across the evolution of these two species, with *P. pacificus* acquiring distinct receptor expression throughout many of its head neurons including the IL2 sensory neurons. Concurrent studies have revealed the IL2 neurons in *P. pacificus* also express both mechanosensory and chemosensory receptors necessary for efficient prey detection, indicating a convergence of these processes into the same neuronal circuitry[60]. Crucially, the lineage-specific noradrenergic adaptations parallel the evolution of predatory behaviour in the Diplogastridae, revealing an ancient origin for this association. Thus, noradrenergic circuits balance aggressive behavioural states in predatory nematodes and are associated with the evolution of complex behavioural traits.

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

## Methods

### Animal handling and maintenance

The strains used in this study are shown in Supplementary Table 1. *C. elegans* and *P. pacificus* were maintained at 20 °C on nematode growth medium (NGM) agar plates containing *Escherichia coli* OP50.

### Transgenic animals

To generate the *Ppa-myo-2p*::RFP (JWL27) strain, we used the previously established protocol[61]. This construct was generated by PCR amplification of a 1,231-base pair (bp) upstream region in front of the first predicted ATG start codon of *Ppa-myo-2* and subsequent cloning into the pZH009 containing the codon-optimized red fluorescent protein (TurboRFP) plasmid. NEBuilder HiFi DNA Assembly Master Mix (New England Biolabs) was employed to perform cloning. To generate the transcriptional reporters of *Ppa-tdc-1*, *Ppa-ser-3*, *Ppa-ser-6* and *Ppa-lgc-55*, we cloned the upstream regions before their predicted start codon, including: *Ppa-tdc-1*: 1,585 bp, *Ppa-ser-3*: 1,996 bp, *Ppa-ser-6*: 1,996 bp and *Ppa-lgc-55*: 1,917 bp, to drive expression of the codon-optimized TurboRFP or GFP as required. Each injection mix contained 10 ng μl$^{-1}$ of the PstI-HF-digested reporter plasmids, 10 ng μl$^{-1}$ of the PstI-HF-digested *Ppa-egl-20p*::GFP plasmid as co-injection marker and 60 ng μl$^{-1}$ of the PstI-HF-digested *P. pacificus* genomic carrier DNA. The mix was injected in the gonads of young adults. Between 50 and 3,000 animals were injected depending on the strain. The transgenic animals were screened using an epifluorescence microscope (Axio Zoom V16; Zeiss). The fluorescence images of the transgenic animals were obtained using a Leica SP8 confocal microscope.

### Transgenic line integration

*Ppa-myo-2p*::RFP was integrated into the *P. pacificus* genome as previously described[62]. Briefly, ten NGM plates each containing 20 fluorescent *Ppa-myo-2p*::RFP animals were exposed to ultraviolet irradiation at 0.050 J cm$^{-2}$ using a UVP Crosslinker (CL-3000 Analytik Jena). After 3–4 days F1 fluorescent animals were singled out onto 120 individual culture plates and after another 3–4 days the F2 progeny were screened for possible integration events. This was detected by observing an increase in the number of fluorescent animals to ≥75% of the population. Individual animals from these plates were isolated and screened for consistent 100% transmission. Integrated lines were subsequently outcrossed 4× to remove potential mutations caused by ultraviolet exposure.

### Behavioural imaging

*Ppa-myo-2p*::RFP animals were recorded at ×1 effective magnification using an epifluorescence microscope (Axio Zoom V16; Zeiss) as previously described[19]. Recordings were made through a Basler camera (acA3088-57um; BASLER) with 15-ms exposure time. Animals were imaged at 30 frames per second for 10 min unless otherwise indicated. All animals that were in the field-of-view for at least 60 s were included in the analysis.

For tracking of animals on predatory assays, *C. elegans* prey were first maintained on OP50 bacteria until freshly starved, resulting in an abundance of young larvae. These plates were washed with M9, passed through two 20-μm filters, centrifuged and deposited onto the assay plate by pipetting 4 μl of worm pellet onto a 6-cm NGM unseeded plate. A copper arena (1.5 × 1.5 cm$^2$) was placed in the middle of the assay plate to constrain predators in the recording field. Forty young adult *P. pacificus* predators (eurystomatous mouth form) were starved for 2 h and then added to assay plates inside the arena. After a recovery period of 15 min on the assay plate, their behaviours were recorded for 10 min.

For tracking of animals on bacterial assays, 300 μl of *E. coli* OP50 overnight culture was spotted onto an empty 6-cm NGM plate 24 h before the assay. A copper arena (1.5 × 1.5 cm$^2$) was placed in the middle of the assay plate to contain the *P. pacificus* in the recording field. Forty young adult predators were starved for 2 h and then added to assay plates inside the arena. After a recovery period of 15 min on the assay plate their behaviours were recorded for 10 min.

### Automated behavioural tracking

Animals were tracked using the custom Python analysis package 'PharaGlow'[19]. PharaGlow performs a three-step analysis: (1) centre of mass (CMS) tracking and collision detection; (2) linking detected objects to trajectories; and (3) extracting centreline, contour, width and other parameters of the shape to allow extracting pharyngeal pumping events. The generated files contain the position and the straightened images which are further processed to extract the behavioural measures. Post-processing was applied according to ref. 19 with minor modifications. To obtain pumping traces from straightened animals, the inverted skew of intensity is calculated for each frame per animal (Fig. 1c). This metric is sensitive to the opening of the pharyngeal lumen and pharyngeal contractions. Peaks in the resulting trace correspond to pumping events.

### Feature engineering

Videos of animals with labelled pharyngeal muscles were processed using PharaGlow[19], which provided a set of initial behavioural features (CMS coordinates, the centreline, the pumping rate and the skew of the fluorescence intensity distribution, which relates to pumping contractions). From these basic features, we calculated two derived features. Using the CMS coordinates $(x(t), y(t))$, velocity was calculated with a time shift of d$t$ = 60 frames (2 s). To obtain a description of head motion, we calculated the angle between the movement direction of the CMS and nose tip as

$$\theta = \arccos(\mathbf{v}_{CL}\mathbf{v}_{CMS})$$

where $\mathbf{v}_{CL}$ is the unit vector of the centreline between the 1st and 5th coordinates along the 100 equidistantly sampled points of the centreline and $\mathbf{v}_{CMS}$ is the unit vector of the CMS coordinates between two consecutive frames. To capture frequency changes in these features, we apply a wavelet transform using pywt with a gaus5 wavelet and extract the maximum frequency for each feature over time. The head angle and the skew which relates to the faster pumping motion were transformed with the following range of pseudo-frequencies (0.3–5.0 Hz). From the wavelet transformations the maximum frequency was extracted and included as a feature. Only for skew, two wavelet transforms from scale 11.2 and 3.8 were directly included additionally, which translates to pseudo-frequencies of about 1.3 and about 3.9 Hz, frequencies relevant for feeding behaviour. For velocity only one wavelet transform with the scale 18.7 was included, which translates to around 0.8 Hz. This transformation resulted in a total set of nine features.

### Data curation

During recordings with many animals, a small subset of frames include overlapping animals. We detect and remove these instances in the data by removing frames with an area higher than 1.5 times the mean area in one recording.

### Manual annotation

An expert human annotator generated labels for a subset of frames following the naming conventions for behaviour in ref. 16. Using Label-Studio pumping rate and velocity data were shown, and the annotator marked sequences corresponding to 'biting', 'feeding', 'exploration' or 'quiescence'. The labels were verified by playing the corresponding video of the animal. For the unlabelled recordings, we automatically set labels encoding the recording condition, namely 'on OP50' or 'on *C. elegans* larvae'. For preprocessing, the original features were preprocessed and downsampled before analysis using the following pipeline

implemented in sklearn. First, further lagged features were created from the base features (velocity, instantaneous pumping rate, head angle, mean pumping rate) by shifting the individual dataset by 5, 10 and 15 frames (with 30 frames per second) forward or backward in time. This allowed us to include a short history in an otherwise memory-less pipeline. Subsequently, features were averaged using a rolling window of 30 frames (1 s) and downsampled to 1 frame per second. Labels were downsampled by using the modal value within 30 frames (1 s). The first and last seconds of each recording were truncated to eliminate effects of smoothing artifacts. The data were then transformed with a Yeo–Johnson power transform to improve normality and normalized with robust scaling. The entire pipeline was fitted on the training dataset and also applied during prediction of new data. For dimensionality reduction and clustering, the preprocessed training data with $n = 106$ animals from WT recordings on larvae and bacteria were embedded in three dimensions using UMAP (umap module, Python), using the parameters: n_neighbors = 70, min_dist = 0, repulsion_strength = 4, negative_sample_rate = 15, disconnection_distance = 0.85, n_components = 3. The resulting embedding was clustered with HDBSCAN. The number of clusters was determined using a silhouette score. To assign human-interpretable behavioural states to the clusters, the overlap between clusters and manual labels was calculated (Extended Data Fig. 2c) and cluster labels were verified by inspecting the feature distributions within the different clusters. Finally, cluster labels were spot-checked using the video data.

### Behavioural state classification

Behavioural feature data with cluster labels found by embedding and clustering were used to train an XGBoost Classifier on the preprocessed data. Of the 106 videos, 9 were held-out as a test set and were not used for training the classifier. These videos were selected for their similarity in label distribution compared with the training set. Bayesian optimization was used to find optimal hyperparameters for the classifier using cross-validation. Training data were split using a stratified group shuffle split, whereby each recording was constituting a group. After the best hyperparameters were found, a model was trained on the complete training data using the optimal hyperparameters. Next, the performance of the model was evaluated with the prediction of the test set in comparison with the cluster labels. For this, standard model performance metrics were calculated (Fig. 1 and Extended Data Fig. 3a). Before prediction, the steps feature engineering and preprocessing are applied to the new data. To ensure reliability of the predictions, predictions with a probability of less than 50% were set to 'None'. This threshold balances the rate of false-positive detections with the number of unlabelled frames. In our case, excepting frames at the start and end of each recording which were excluded because of filtering effects, this threshold resulted in only 1.2% of all frames classified as 'None'.

### Cluster validation

To validate the robustness of the identified behavioural states, the same pipeline was applied to an independent dataset, with $n = 254$ animals. The dataset was resampled to contain the same number of frames from worms recorded on larvae and on OP50 as the original training set. For dimensionality reduction the same parameters were used as previously described.

### Dual-colour epifluorescence microscope behavioural tracking

To investigate predator–prey interaction, we recorded *P. pacificus* predators expressing the *Ppa-myo-2p*::RFP (JWL27) pharyngeal marker interacting with *C. elegans* larvae expressing *Cel-myo-3p*::GFP (SJ4103). Freshly starved plates of *Cel-myo-3p*::GFP prey were washed with M9, filtered two times with a 20-µm nylon filter and centrifuged at 200 rpm to extract a pure larval culture. Then, 0.5 µl of the pellet was transferred onto a 6-cm NGM imaging plate. The larvae were allowed to disperse undisturbed for at least 1.5 h. *P. pacificus Ppa-myo-2p*::RFP young adults

were starved for 2 h and subsequently transferred onto the imaging plates. Worms were recorded with a dual-colour epifluorescence tracking microscope[29]. The microscope featured a 16-mm objective and a 50-mm tube lens, resulting in a magnification of approximately ×3.1. The camera used was a Basler acA3088-57um. To compensate for the different brightness of the fluorophores, the red channel was attenuated by a neutral density filter with an optical density of 0.6. Recordings of individual predators were analysed using a similar approach to the large population recordings with an adaptation to account for the stage motion. Prey signals were extracted from the GFP channel with a 22-µm-wide circular mask, centred at the anterior end of the predator using the extracted centreline of the pharynx as a guide. Prey signals was calculated as:

$$\text{Prey signal} = \frac{(\text{GFP}_{95\,\text{percentile}} - \text{GFP}_{5\,\text{percentile}})}{\text{GFP}_{5\,\text{percentile}}}$$

Tracks were aligned to the onset of biting events, and the prey signal was normalized to the baseline, defined as 15 to 5 s before bite onset:

$$\text{Prey signal} \% = \frac{(\text{Prey signal} - \text{Baseline})}{\text{Baseline}}$$

Statistics were performed using an upper-tailed paired *t*-test, comparing the mean prey signal (%) of 15 to 5 s before bite onset and 0 to 15 s. To investigate the ingestion of GFP-labelled body wall muscle by the predator, we analysed the signal of the green channel subtracted (background subtracted with the 'rolling ball' algorithm in ImageJ with a 10-px (7.4 µm) radius). A kymograph was extracted along the centreline of the pharynx using ImageJ for one exemplary video. The video was selected because the predator was stationary for a prolonged time of 30 s, whereas the focal plane clearly showed ingestion of labelled food. The linewidth of the kymograph was 6 px (4.4 µm). To detect individual ingestion events, peak detection was performed using a segment of the kymograph in the range of the metacorpus where the ingested food was most easily visible. The intensity values in this region were averaged for each time point.

### Statistical analysis

For the statistical analysis of the state predictions (relative time in state, mean bout duration, transition rates), and for the corpse assay, the two-tailed Mann–Whitney *U*-test was used. The statistical analysis was performed on each condition and state with its respective control. A Bonferroni correction was applied to take into account that the probability of a false-positive significant test rises with the number of compared conditions. Thus, the *P* values reported here are corrected for the number of comparisons made with the respective control. All raw *P* values and sample sizes are available in Supplementary Table 3. The box plots follow Tukey's rule for which the middle line indicates the median, the box denotes the first and third quartiles, and the whiskers show the 1.5 interquartile range above and below the box. In all figures, * denotes a significance value between 0.05 and 0.01, ** a significance value between 0.01 and 0.001, *** a significance value between 0.001 and 0.0001, and **** a significance below 0.0001.

### Generation of CRISPR–Cas9-induced mutations

The procedure for CRISPR–Cas9 mutagenesis was based on previously described *P. pacificus* methods[63]. Briefly, gene-specific CRISPR RNAs were designed to target early predicted exons in all target genes and synthesized by Integrated DNA Technologies (IDT). These were then fused to tracrRNA (IDT) at 95 °C for 5 min before cooling to room temperature and annealing. The hybridization result was coupled with purified Cas9 protein (IDT). After 5 min of incubation at room temperature, Tris-EDTA buffer was added to achieve a final concentration of 18.1 µM for single guide RNA (sgRNA) and 12.5 µM for Cas9. The mix

was injected in the gonads of young adults. P0 worms were discarded 12–24 h after microinjection. After 2 days, 96 F1 progeny from P0 plates were singled out and allowed to lay eggs for 24 h. The genotypes of the F1 animals were subsequently analysed through Sanger sequencing. For detailed information about the mutants created during this study, see Extended Data Fig. 9 and Supplementary Table 1. All sgRNAs and the primers used in this study to generate mutants can be found in Supplementary Table 2.

## Pharmacological experiment

To quantify the effects of exogenous octopamine and tyramine, predatory feeding assays were performed on agar plates which were supplemented with 2 mM octopamine and tyramine, respectively. Tyramine and octopamine plates were prepared by adding tyramine-HCl (Sigma) and octopamine-HCl (Sigma) to a concentration of 2 mM to freshly autoclaved NGM that was cooled to 55 °C before use. *Ppa-myo-2p*::RFP animals were picked onto either unseeded octopamine- or tyramine-containing plates and starved for 2 h. These animals were subsequently added to standard assay plates with *C. elegans* larvae contained in a copper arena and also containing octopamine or tyramine[12].

## Egg-laying assay

Three-day-old adult worms were placed on individual plates seeded with OP50. The following day, the number of eggs was manually counted and the worms were transferred onto new plates. The eggs were counted for 5 days per individual worm after which only unfertilized eggs were laid.

## Manual predation assays

Assays were conducted as previously reported[16]. Briefly, prey were maintained on NGM plates seeded with OP50 bacteria until freshly starved, resulting in an abundance of young larvae. These plates were washed with M9 and passed through two 20-μm filters to isolate larvae. Then, 1.0 μl of *C. elegans* larval pellet was transferred onto a 6-cm unseeded NGM plate. Five predatory nematodes were screened for the appropriate mouth morph and added to assay plates for prey assays. After 2 h the plate was screened and corpses were counted manually.

## Worm size measurement

Synchronized J2 larvae were placed onto NGM plates with bacteria. They were transferred from assay plates to NGM plates without bacteria during different developmental points (24, 48 and 72 h). Bright-field images of the worms were taken using an epifluorescence microscope (Axio Zoom V16; Zeiss) and the Basler camera (acA3088-57um; BASLER). Images were analysed using the Wormsizer plugin for ImageJ/Fiji. Wormsizer detects and measures the area of the worms.

## Mouth-form phenotyping

Mouth-form phenotyping was performed as previously reported[17]. In brief, NGM plates with synchronized young adults were placed onto a stereomicroscope with high magnification (×150). Then, 100 worms were screened for the mouth form per strain. The eurystomatous mouth form was determined by the presence of a wide mouth, whereas the stenostomatous forms were determined by a narrow mouth. Eurystomatous young adult worms were picked for predation assays. Images of mouth forms were taken using a differential interference contrast (DIC) ×60 lens with a ×1.6 magnifier.

## *Ppa-myo-2p*::RFP copy number quantification

Thirty J4/young adult hermaphrodites from a non-starved plate were picked for DNA extraction using a NEB Monarch DNA kit (final elution volume 35 μl). A whole plate of non-starved worms was used for RNA extraction using a Zymo RNA mini kit (final elution volume 30 μl). To determine the copy number of *Ppa-myo-2p*::RFP in the ultraviolet integrated transgenic line (JWL27) and relative expression of this construct,

quantitative PCR (qPCR) and qPCR with reverse transcription assays were conducted. For copy number, primers were designed to amplify a part of the RFP sequence and compared with two known single-copy gene sequences, *Ppa-gpd-3* and *Ppa-csg-1*. As *Ppa-myo-2p*::RFP is a transcriptional reporter, no part of the *Ppa-myo-2* gene sequence is included in the reporter line. Therefore, to determine the relative expression of *Ppa-myo-2p*::RFP, the expression of RFP was compared with that of the native *Ppa-myo-2* exon gene sequence. qPCR primers specific for RFP were: forward GGAGAGGGAAAGCCTTACGAGG and reverse GAATCCC TCAGGGAAAGACTGC. Two pairs of gene-specific *Ppa-myo-2* primers were used. These were pair 1: forward CGAAGAAGAACGTGTGGGTG and reverse TACCTCATTGCCGGGACCTC; and pair 2: forward AGGAG ACAAAGGGAGACACG and reverse GGGTTCATCTCCTGCACTTGG. For quantification, DNA samples were 1:10 diluted with water, whereas RNA samples were not diluted. Three technical replicates and two biological replicates were conducted.

## HCR

HCR RNA fluorescence in situ hybridization was performed in *P. pacificus* using a modified version of the protocol described previously[64]. The protocol was adjusted to include an extended Proteinase K treatment step (final concentration of 200 μg ml⁻¹ for approximately 1 h at room temperature) to improve tissue permeability and probe penetration. Worms were incubated overnight at 37 °C in probe hybridization buffer containing 200 pmol of each probe set. We examined gene expression using dual and triple HCR labelling. The genes *Ppa-tbh-1* labelled with X1-488 and *Ppa-tdc-1* labelled with X2-647 were imaged together. Similarly, *Ppa-klp-6* labelled with X1-488 and *Ppa-ser-3* labelled with X2-647 were co-detected in the same specimens. Triple labelling was carried out to visualize *Ppa-ser-3* using B2 (Alexa Fluor 546) probes, *Ppa-ser-6* using *Ppa-ser-6* to B3 (Alexa Fluor 488) probes and *Ppa-lgc-55* using B1 (Alexa Fluor 647) probes.

## Generation of HisCl transgenic strains

To silence neurons in *P. pacificus*, we used a codon-optimized version of the *C. elegans* histamine-gated chloride channel (HisCl1), following the strategy described previously[65,66]. For RIM and RIC neuronal silencing, the codon-optimized HisCl1 sequence was inserted downstream of the *Ppa-tdc-1* promoter (1,585 bp) to generate the *Ppa-tdc-1p*::HisCl plasmid. For IL2 silencing, the *Ppa-klp-6* promoter (2 kilobases) was used to generate the *Ppa-klp-6p*::HisCl plasmid. For microinjections, the *Ppa-tdc-1p*::HisCl injection mix contained 10 ng μl⁻¹ of SalI-HF-digested *Ppa-tdc-1p*::HisCl plasmid, 10 ng μl⁻¹ of PstI-HF-digested *Ppa-mec-6p*::Venus plasmid as a co-injection marker and 60 ng μl⁻¹ of PstI-HF-digested genomic carrier DNA. For the *Ppa-klp-6p*::HisCl injection mix, this contained 10 ng μl⁻¹ of PstI-HF-digested *Ppa-klp-6p*::HisCl plasmid, 10 ng μl⁻¹ of PstI-HF-digested *Ppa-klp-6p*::GFP plasmid as a co-injection marker and 60 ng μl⁻¹ of PstI-HF-digested genomic carrier DNA. Transgenic animals were identified by GFP or Venus expression and maintained as extrachromosomal lines for subsequent behavioural analysis.

## Preparation of histamine-containing assay plates

To induce neuronal silencing through activation of HisCl1 channels, NGM was supplemented with 10 mM histamine dihydrochloride, as described previously[65,66]. The 1 M histamine stock solution was prepared in sterile distilled water and added to molten NGM cooled to approximately 60 °C at a volume of 5 ml per 500 ml of agar. Plates were poured, allowed to solidify at room temperature and used within 1 week. These histamine plates were used both during the 2-h starvation period before behavioural assays and during the assays themselves.

## Behavioural assays for neuron silencing

Behavioural assays were conducted post silencing to examine the roles of RIC and RIM neurons for functional octopamine and

tyramine release, as well as the importance of the IL2 neurons for predatory aggression. For IL2 neuron silencing, 40 young adult *Ppa-klp-6p*::HisCl animals (eurystomatous morph) were starved for 2 h on 10 mM histamine plates and subsequently transferred to an assay arena consisting of a 10 mM histamine plate containing *C. elegans* L1 larvae as prey. After a 15-min recovery period, predator behaviour was recorded for 10 min. To silence RIM and RIC neurons, *Ppa-tdc-1p*::HisCl animals in a *tbh-1* mutant background were used. Animals were placed on 10 mM histamine plates seeded with OP50 bacteria overnight before the assay to ensure prolonged silencing and minimize any residual tyramine release from RIM or RIC neurons. Prey assay conditions were otherwise identical to those described for IL2 silencing.

### Gene phylogenetic analysis

The predicted evolutionary history of the neuromodulator biosynthesis enzymes and associated receptors was determined by reciprocal best BLAST matches with gene predictions in genome assemblies for *P. pacificus* and *C. elegans*. Phylogenetic inference was conducted on the amino acid sequence using the Maximum Likelihood criterion and a Jones–Taylor–Thornton matrix-based model. The tree with the highest log likelihood is shown. Initial tree(s) for the heuristic search were obtained automatically by applying Neighbor-Join and BioNJ algorithms to a matrix of pairwise distances estimated using the Jones–Taylor–Thornton model, and then selecting the topology with superior log likelihood value. The tree is drawn to scale, with branch lengths measured in the number of substitutions per site. Evolutionary analyses were conducted in MEGA X[67].

### Nematode species phylogeny

A species phylogeny was adapted from previous work[68]. The original tree was redrawn and simplified to show only the branching order (topology) and emphasize species relationships. Branch lengths were omitted, as relative divergence times are not relevant for the present analysis.

### Reporting summary

Further information on research design is available in the Nature Portfolio Reporting Summary linked to this article.

### Data availability

All data are available in the main text, Supplementary Tables 1–3 and Supplementary Videos 1–5, or as tracking data on OSF at https://osf.io/cua87. Raw data from behavioural tracking will be made available via file transfer upon reasonable request, as these data are more than 200 GB.

### Code availability

The code underlying the data analysis presented in this manuscript can be accessed under a GPL 3.0 license at https://github.com/scholz-lab/PpaPred.

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

**Acknowledgements** We thank R. J. Sommer, O. Hobert, C. Rödelsperger and R. Hong for useful discussions, W. Bönigk for transgenic cloning assistance (Genetics Facility, MPI for Neurobiology of Behavior – caesar) and J. Berger for SEM imaging (MPI for Biology). Additionally, we thank the Sommer lab for *P. pacificus* and *A. Sudhausi* wild-type strains (MPI for Biology) and S. Suo for *C. elegans* strain VN443 (Saitama Medical University). Finally, the *C. elegans* wild-type N2 strain, SJ4103, BC11545 and FQ63 were provided by the Caenorhabditis Genetics Center (CGC), which is funded by the NIH Office of Research Infrastructure Programs (grant no. P40 OD010440). This work was funded by the Max Planck Society, by the German Research Foundation (DFG) project number 495445600 and by the iBehave Network (sponsored by the Ministry of Culture and Science of the State of North Rhine-Westphalia). Part of this work was funded through the BABots project. The BABots project has received funding from the Horizon Europe, PathFinder European Innovation Council Work Programme under grant agreement no. 101098722.

**Author contributions** Conceptualization: G.G.E., L.B., M.S. and J.W.L. Methodology: G.G.E., L.B., M.S. and J.W.L. Investigation: G.G.E., L.B., M.R., F.H., J.L., L.A., D.L.G., L.A.C., N.Z., Z.H., M.O., M.S. and J.W.L. Visualization: G.G.E., L.B. and M.S. Funding acquisition: M.S. and J.W.L. Project administration: M.S. and J.W.L. Supervision: M.S. and J.W.L. Writing—original draft: M.S. and J.W.L. Writing—review and editing: G.G.E., L.B., D.L.G., M.S. and J.W.L.

**Funding** Open access funding provided by Max Planck Society.

**Competing interests** The authors declare no competing interests.

**Additional information**
**Correspondence and requests for materials** should be addressed to Monika Scholz or James W. Lightfoot.

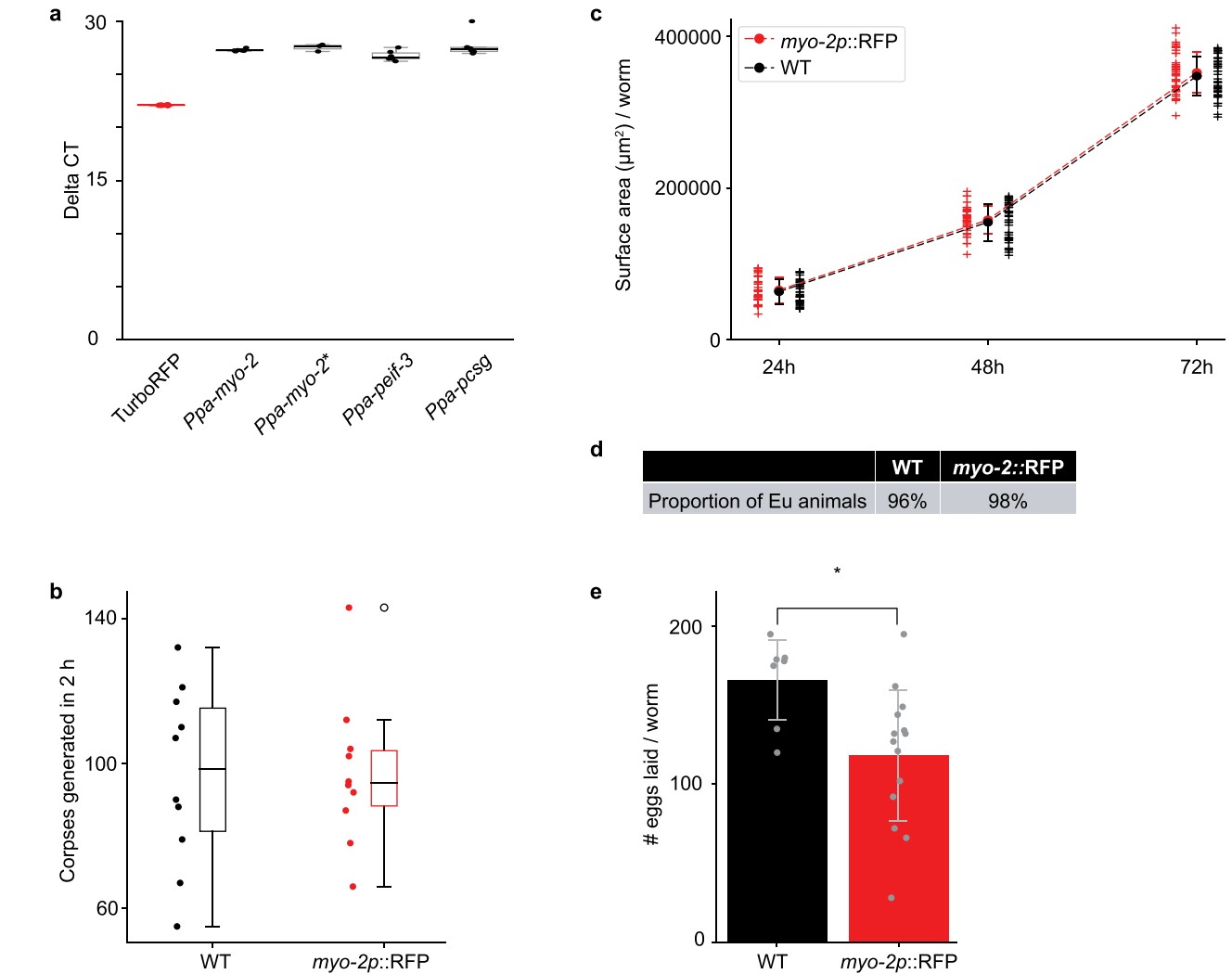

**Extended Data Fig. 1 | Integration of *Ppa-myo-2p*::RFP.** (**a**) qPCR analysis of RFP copy number. RFP qPCR reveals 5.5 less cycles of RFP compared to two different primer pairs (second primer denoted with *) specific to the *Ppa-myo-2* gene and two known single copy genes *Ppa-csg-1* and *Ppa-eif-3* indicating ~32-45 copies of *Ppa-myo-2*::RFP are integrated into the *P. pacificus* genome in strain JWL27. n = 6 (**b**) Predatory behaviour is unaffected in the *Ppa-myo-2p*::RFP integrated line. Standard corpse assays with 5 young adult *P. pacificus* predators placed onto assay plates containing an abundance of *C. elegans* larvae to predate on for 2 h. 10 replicates were conducted for wild type and *Ppa-myo-2p*::RFP. (**c**) Growth rate and (**d**) Summary table of mouth form ratios. The phenotypically plastic mouth morph frequencies are unaffected in the *Ppa-myo-2p*::RFP integration line. Eurystomatous (Eu) is the predatory mouth form in *P. pacificus*. (**e**) There is a small reduction in fecundity associated with the *Ppa-myo-2p*::RFP integration. Significance was assessed using a Mann-Whitney U-test (p = 0.015).

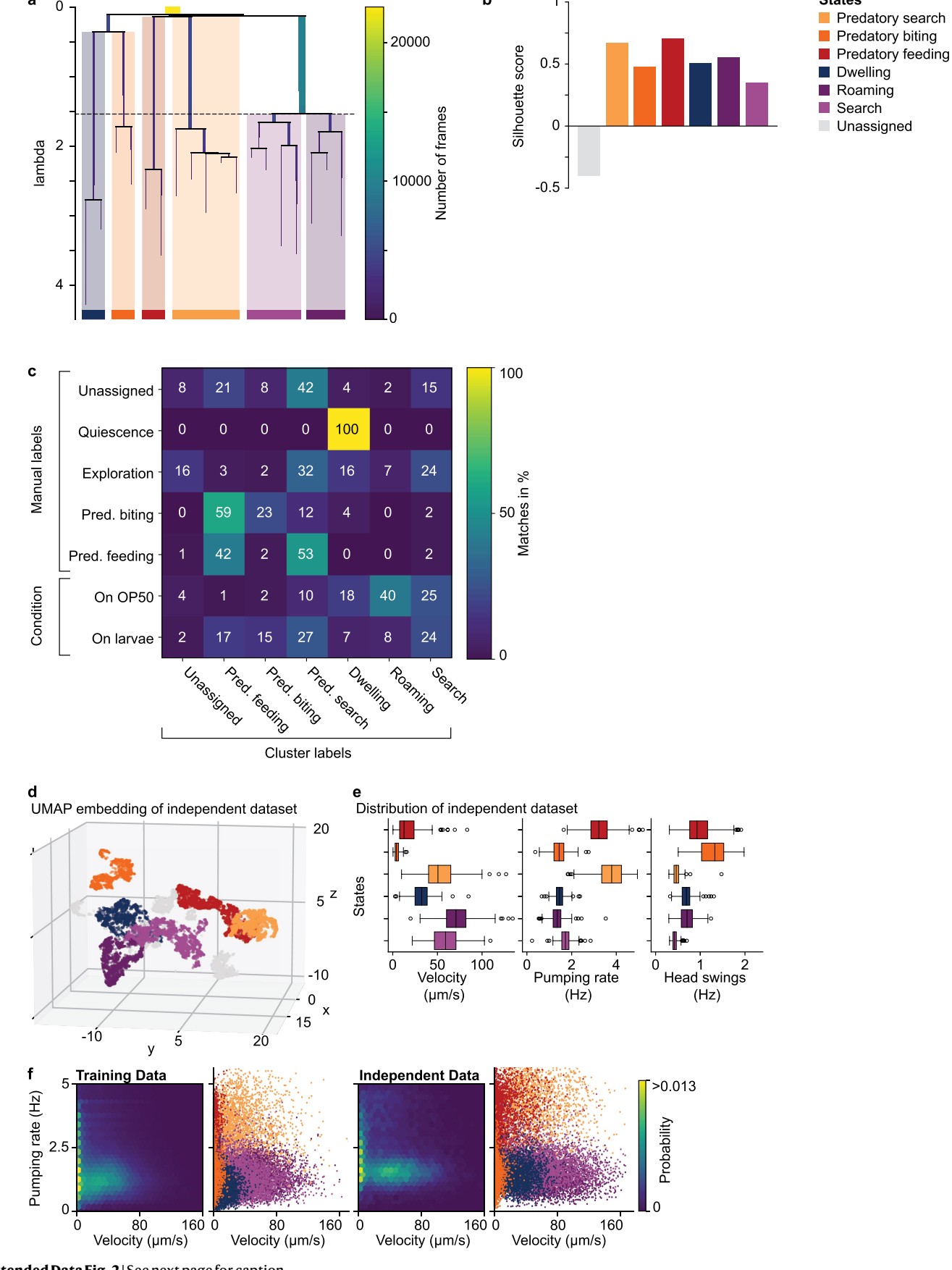

**Extended Data Fig. 2** | See next page for caption.

**Extended Data Fig. 2 | Clustering of behavioural data and state identification.**
(**a**) Hierarchical clustering of UMAP-embedded behavioural features. Colors correspond to the individual clusters that are later identified as behavioural states. The horizontal line indicates where the hierarchical tree was cut. (**b**) Silhouette scores for each cluster. (**c**) Confusion matrix between the human expert annotator, condition, and the cluster labels. Unlabeled data is included by condition as either 'On larvae' or 'On OP50'. The latter was used to identify condition-specific states. (**d**) Cluster validation on a second independent dataset using the same set of parameters for dimensionality reduction. UMAP embedding of behavioural features. Colors indicate the six behavioural states recapitulated by hierarchical clustering. (**e**) Distribution of key behavioural features in each state using a second independent dataset. Each point in the box plots corresponds to the mean value per state and per tracked animal. Box plots follow Tukey's rule with the box from first to third quartiles, and a line at the median. The fliers denote 1.5 x interquartile range. (**f**) Probability density map of velocity and pumping rate for the original dataset used to cluster the behavioural states (left) and an independent dataset to confirm robustness of the state detection.

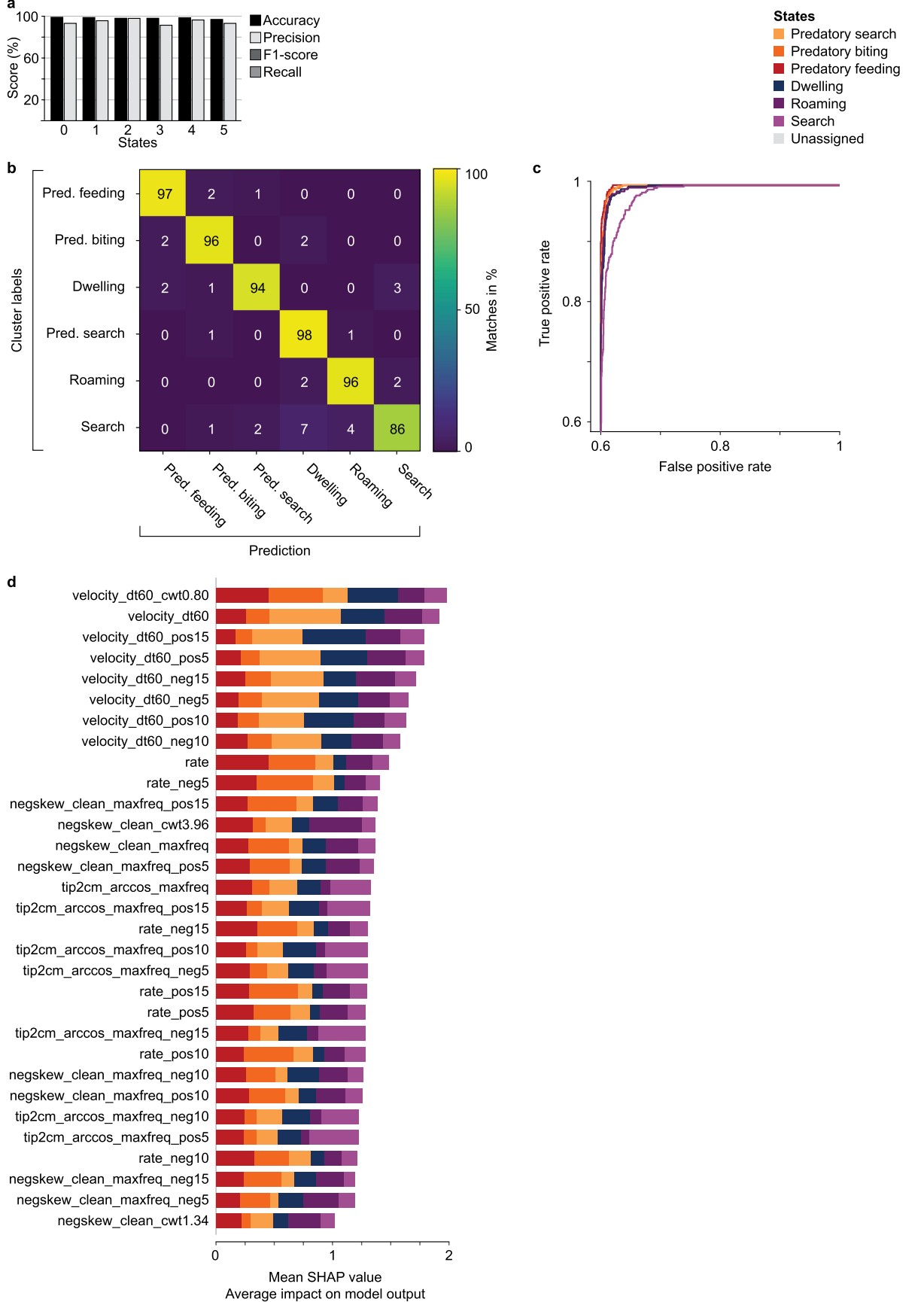

**Extended Data Fig. 3 | Evaluation of the XGBoost classifier model. (a)** Accuracy, precision, recall, and F1-score per state. **(b)** Confusion matrix comparing the cluster labels and the predicted labels using the model. **(c)** ROC curve comparing the false-positive versus false-negative rates per state. **(d)** SHAP feature importance analysis for each feature included in the machine-learning pipeline.

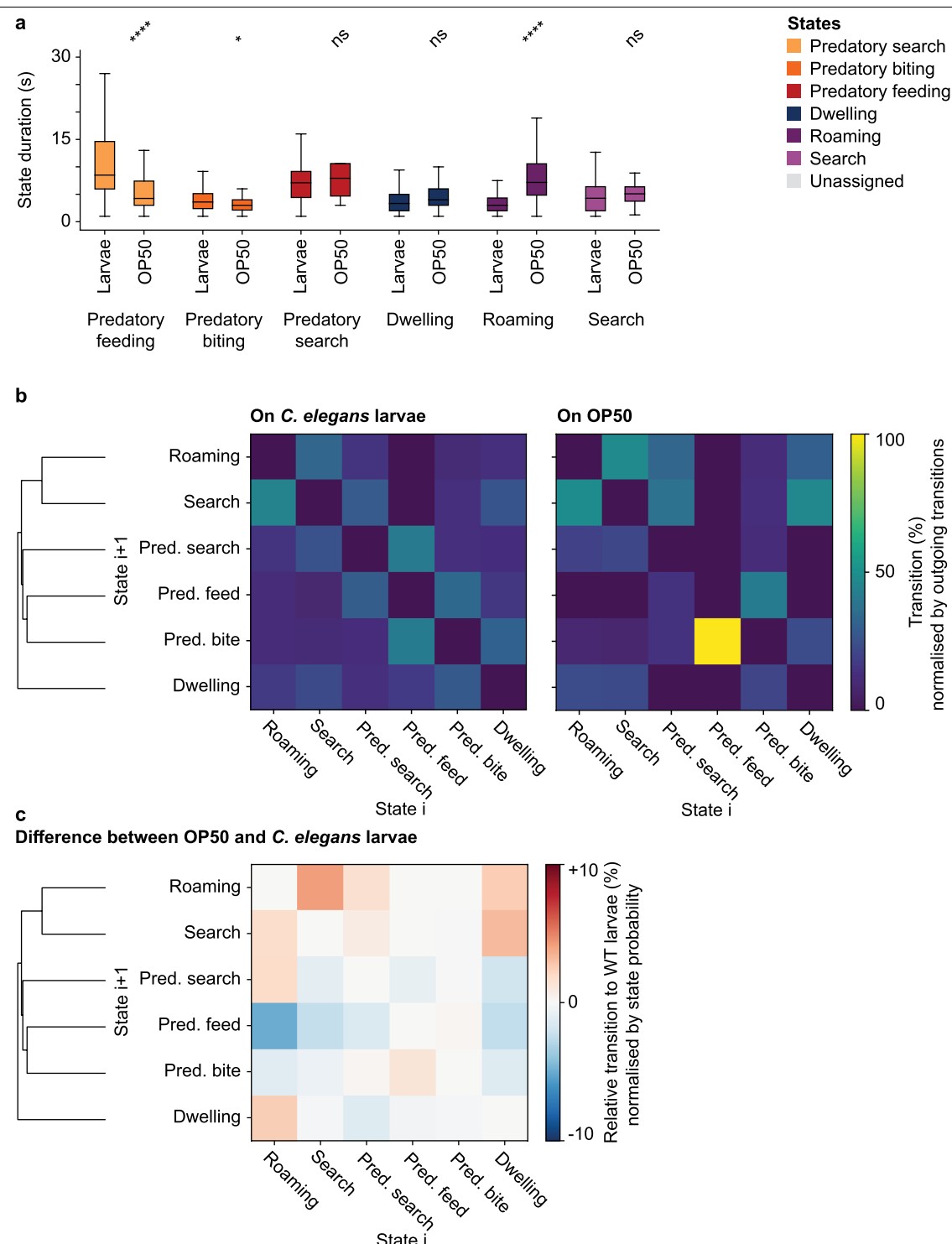

**Extended Data Fig. 4 | Behavioural state prediction and aggression validation for animals in predatory or a bacterial food context.** (**a**) Average state duration for animals on OP50 or on prey larvae. Statistics were performed with Mann-Whitney-U-test using Bonferroni correction for multiple tests against WT on larvae. *p < 0.05, **p < 0.01, ***p < 0.001, ****p < 0.0001; n = 93, n = 99, n = 108, n = 104, n = 79, n = 109 for pred. feed, pred. bite, pred. search, dwelling, roaming, search for WT on larvae respectively and n = 4, n = 24, n = 53, n = 67, n = 86, n = 84 for pred. feed, pred. bite, pred. search, dwelling, roaming, search for WT on OP50, respectively. (**b**) Transition probabilities between states ordered by hierarchical clustering for each food context. (**c**) Difference in transition matrices between the different contexts, scaled by the probability of each state. Note the increase in roaming-dwelling transitions, and the search behaviour on OP50, compared to a reduction of transitions into predatory behaviours like predatory search and predatory feeding.

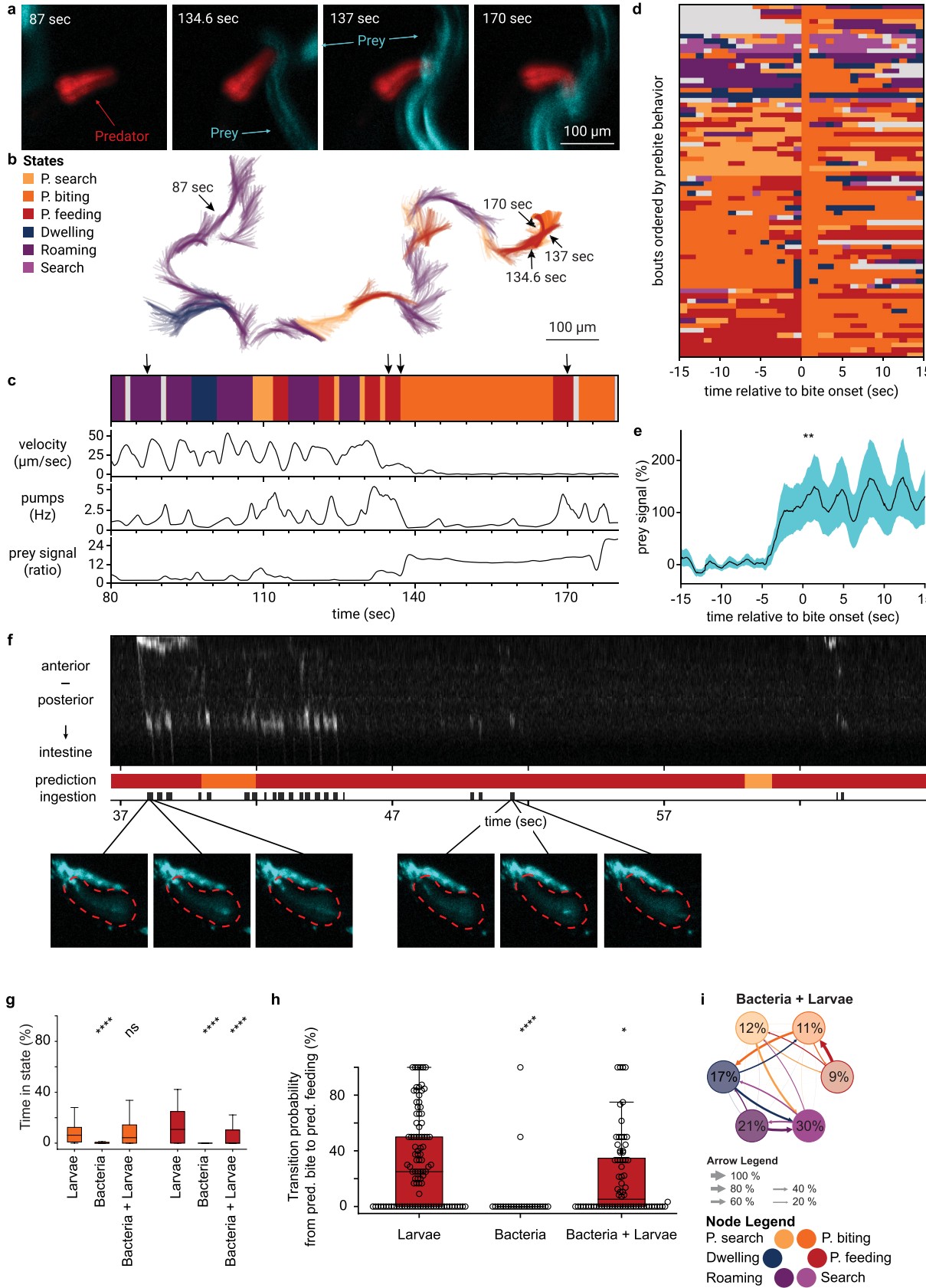

**Extended Data Fig. 5 | Predatory behavioural states correlate with prey contact events.** (**a**) Timeseries of predator-prey contact imaged using dual-color epi-fluorescence tracking microscopy. The pharynx of the predator (red) is tracked as it encounters prey (cyan, body wall labeled). (**b**) Behavioural state prediction overlaid on the x-y track with arrows indicating the times corresponding to the images in (a). (**c**) Ethogram of the behavioural state, the key behavioural features and the fluorescence intensity of the prey for a representative experiment. (**d**) Stacked ethogram for all predicted biting events, sorted by the state predicted prior to biting initiation. (**e**) Average prey signal for all biting events excluding examples where the animal was previously in a biting or predatory feeding state. Statistics were performed with a paired upper-tailed t-test, with N = 10 animals and n = 29 traces. **p < 0.01. (**f**) Kymograph of the prey signal along the pharynx of the predator. Fluorescence indicates ingestion during a sequence of biting and feeding events. The black bars denote individual ingestion events extracted from the kymograph data. Representative images showing parts of the prey animal (cyan) being ingested and swallowed during the predicted feeding state. *P. pacificus* pharynx outlined by red dashed line.

See methods for statistics. (**g**) = Mean fraction of time spent in 'predatory biting' and 'predatory feeding' behavioural states per animal. Box plots follow Tukey's rule with the box from first to third quartiles, and a line at the median. The fliers denote 1.5 x interquartile range. Statistics were performed with Mann-Whitney-U-test using Bonferroni correction for multiple tests against WT on larvae. *p < 0.05, **p < 0.01, ***p < 0.001, ****p < 0.0001; n = 98 for WT on larvae, n = 24 for WT on OP50, n = 72 for WT on OP50+larvae. (**h**) Transition probability that a *P. pacificus* animal transitions from 'predatory biting' to 'predatory feeding' (nutritionatl drive). Analysis conducted with *P. pacificus* surrounded with larval prey, a lawn of bacteria or both bacteria and larval prey. Statistics were performed with Mann-Whitney-U-test using Bonferroni correction for multiple tests against WT on larvae. *p < 0.05, **p < 0.01, ***p < 0.001, ****p<0.0001; n = 131 for WT on larvae, n = 92 for WT on OP50, n = 107 for WT on OP50+larvae. (**i**) Average transition rates between behavioural states for predators surrounded by larval prey and a bacterial food source simultaneously. The number in circles indicates the average state duration and the arrow size indicates the transition rate normalized to outgoing transitions.

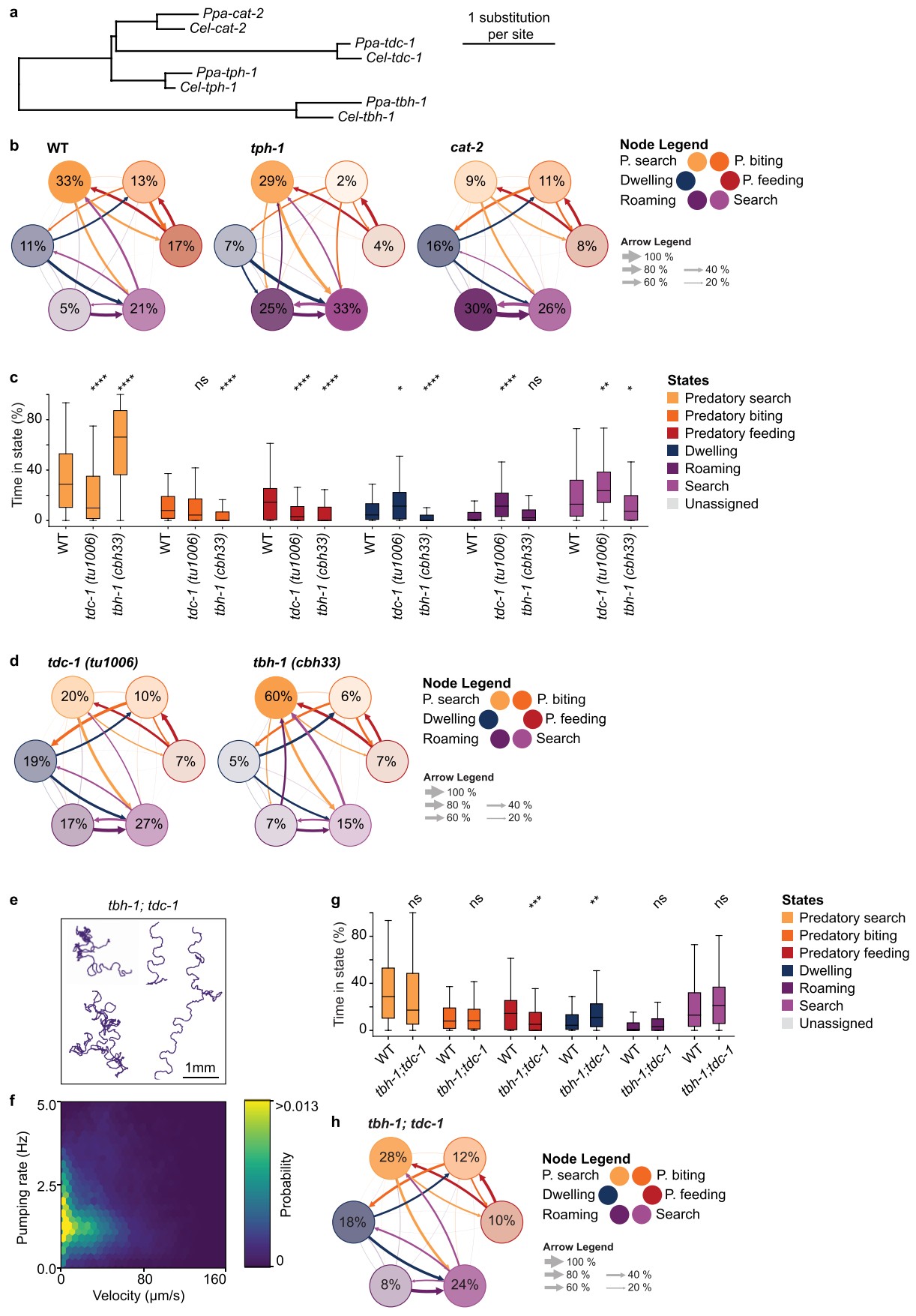

**Extended Data Fig. 6 |** See next page for caption.

**Extended Data Fig. 6 | Behavioural state prediction for *tph-1, cat-2*, and *tbh-1; tdc-1*. (a**) Phylogenetic analysis of genes encoding neuromodulator biosynthesis enzymes from *P. pacificus* and *C. elegans*. The tree supports a single orthologous relationship between species. (**b**) Behavioural state transitions for *tph-1* and *cat-2* mutants. The number in circles indicates the average state duration and the arrow size indicates the transition rate normalized to outgoing transitions. (**c**) Time spent in each behavioural state normalized to the total track duration for a second allele of both *tdc-1*, and *tbh-1*. Box plots follow Tukey's rule with the box from first to third quartiles, and a line at the median. The fliers denote 1.5 x interquartile range. Statistics were performed with Mann-Whitney-U-test using Bonferroni correction for multiple tests against WT. *p < 0.05, **p < 0.01, ***p < 0.001, ****p < 0.0001; n = 91 for WT, n = 124 for *Ppa-tdc-1 (tu1006)*, n = 99 for *Ppa-tbh-1 (cbh33)*. (**d**) Average transition rates between behavioural states for WT, as well as the second alleles of *tdc-1(tu1006)*, and *tbh-1(cbh33)* mutants on larval prey. The number in circles indicates the average state duration and the arrow size indicates the transition rate normalized to outgoing transitions. (**e**) Example tracks and (**f**) probability density map of velocity and pumping rate for the *tbh-1; tdc-1* double mutant. (**g**) Relative time in each behavioural state for WT versus the *tbh-1; tdc-1* double mutant. Statistics were performed with Mann-Whitney-U-test using Bonferroni correction for multiple tests against WT. *p < 0.05, **p < 0.01, ***p < 0.001, ****p < 0.0001; n = 91 for WT, n = 219 for *Ppa-tbh-1-tdc-1*. (**h**) Same as in (d) the average transition rates between behavioural states for the *tbh-1; tdc-1* double mutant.

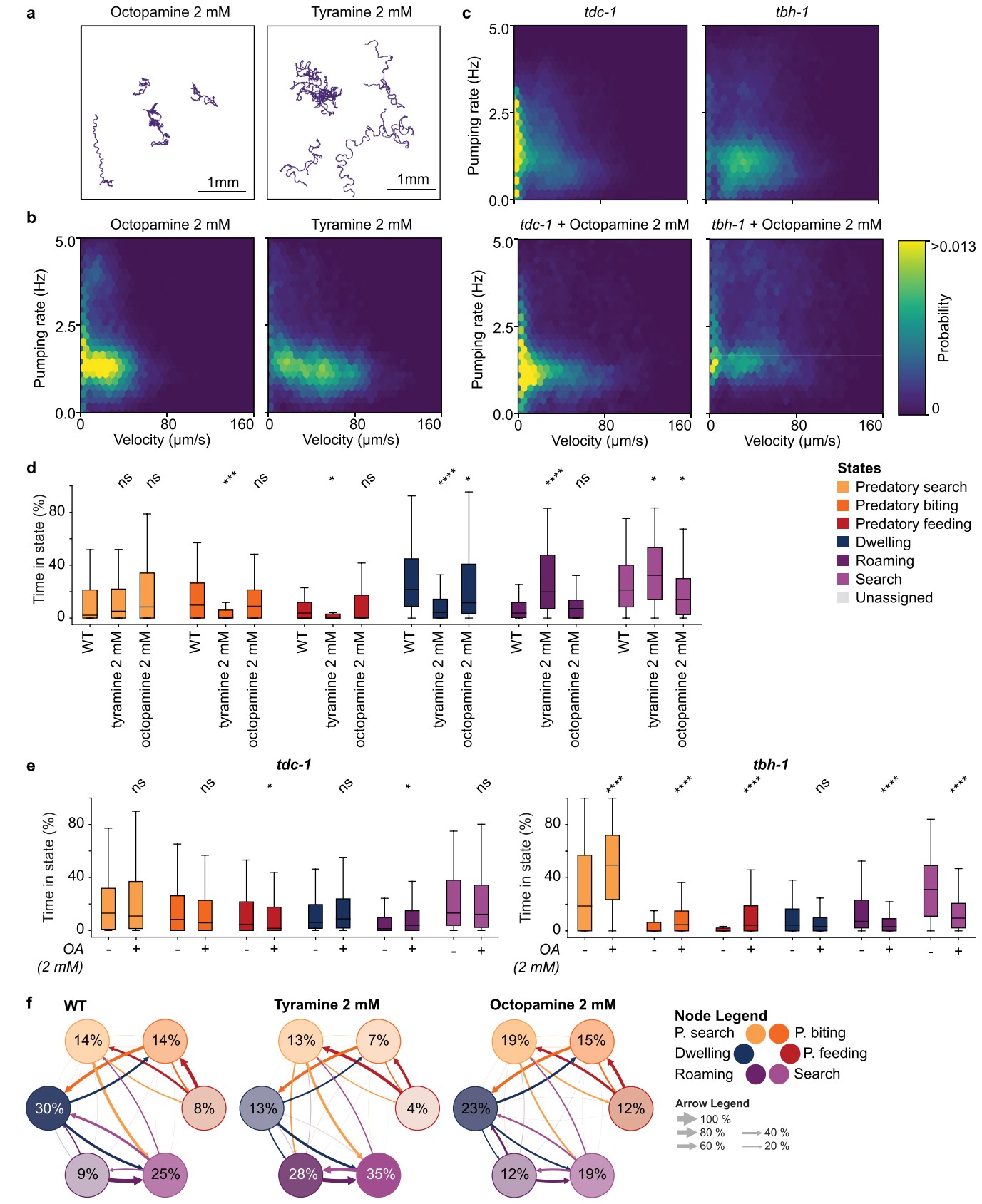

**Extended Data Fig. 7** | See next page for caption.

**Extended Data Fig. 7 | Behavioural state prediction for exogenous application of octopamine and tyramine.** (**a**) Example tracks and (**b**) probability density map of velocity and pumping rate for animals exposed to 2 mM octopamine or tyramine, respectively. (**c**) Probability density map of velocity and pumping rate for *tdc-1* and *tbh-1* mutants exposed to 2 mM octopamine. (**d**) Relative time in each behavioural state for WT animals compared to animals on 2 mM octopamine or tyramine. Statistics were performed with Mann-Whitney-U-test using Bonferroni correction for multiple tests against WT. *p < 0.05, **p < 0.01, ***p < 0.001, ****p < 0.0001; n = 66 for WT, n = 78 for WT treated with Tyramine 2 mM, n = 143 for WT treated with Tyramine 2 mM.

(**e**) Relative time in each behavioural state for *tdc-1* mutants compared to *tdc-1* mutants supplemented with 2 mM exogenous octopamine and *tbh-1* mutants compared to *tbh-1* mutants supplemented with 2 mM exogenous octopamine. Statistics were performed with Mann-Whitney-U-test using Bonferroni correction for multiple tests against *Ppa-tdc-1* (left) and *Ppa-tbh-1* (right). *p < 0.05, **p < 0.01, ***p < 0.001, ****p < 0.0001; n = 235 for *Ppa-tdc-1*, n = 310 for *Ppa-tdc-1* + OA, n = 197 for *Ppa-tbh-1*, n = 401 for *Ppa-tbh-1* + OA. (**f**) Behavioural state transitions for WT, 2 mM tyramine, and 2 mM octopamine. The number in circles indicates the average state duration and the arrow size indicates the transition rate normalized to outgoing transitions.

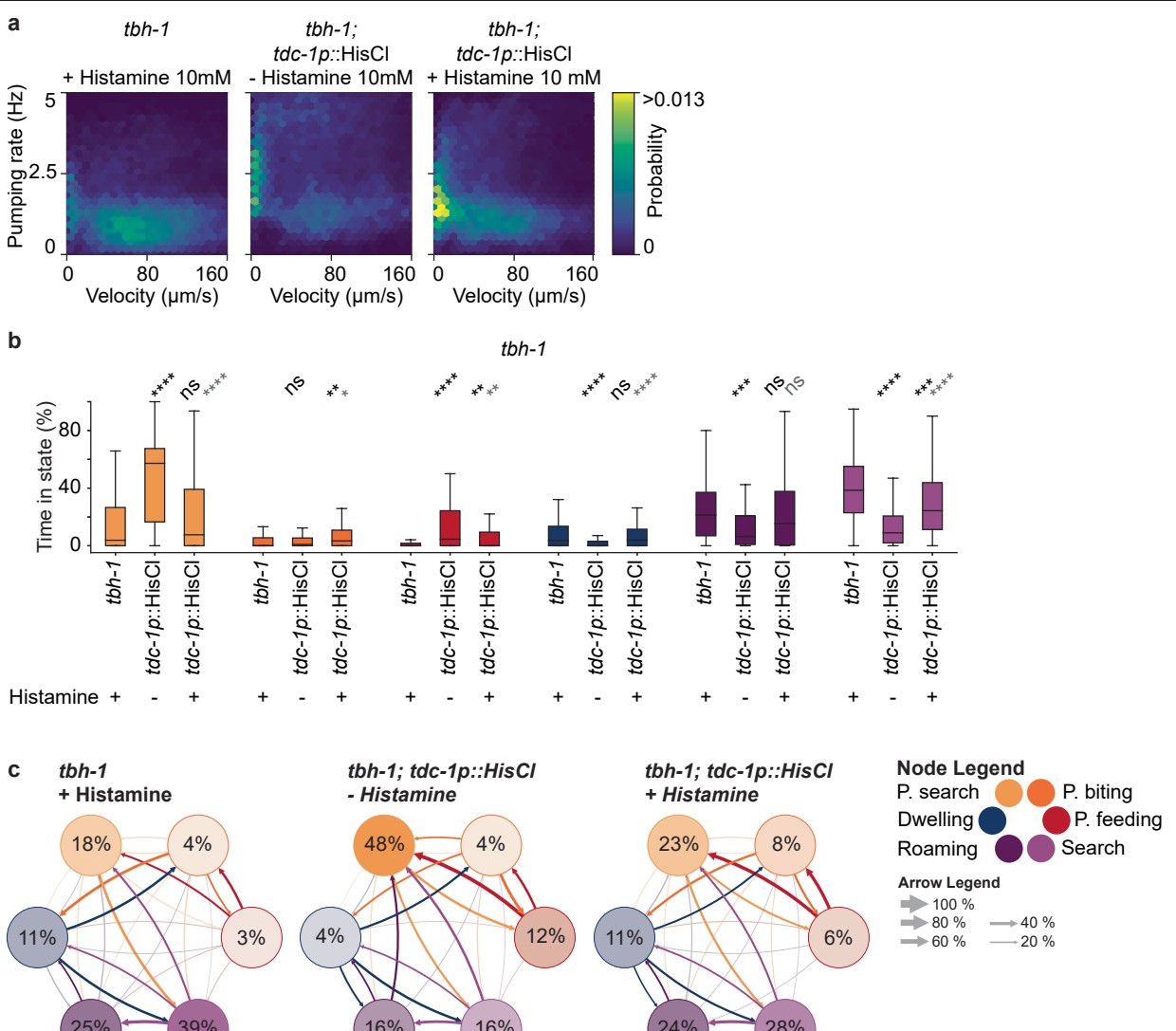

**Extended Data Fig. 8 | Behavioural state prediction assessing the functional site of octopamine and tyramine production.** (**a**) Probability density map of velocity and pumping rate for genetic silencing of RIC and RIM neurons. Plots show *tbh-1* mutant + 10 mM histamine and *tbh-1; tdc-1p*::HisCl without histamine controls and *tbh-1; tdc-1p*::HisCl + 10 mM histamine silencing conditions. (**b**) Relative time in each behavioural state for all conditions in (a). Box plots follow Tukey's rule with the box from first to third quartiles, and a line at the median. The whiskers denote 1.5 x interquartile range. Statistics were performed with Mann-Whitney-U-test using Bonferroni correction for multiple tests against

*Ppa-tbh-1* +Histamine (black asterisks) and *Ppa-tbh-1; Ppa-tdc-1*::HisCl -Histamine (grey asterisks). *p < 0.05, **p < 0.01, ***p < 0.001, ****p < 0.0001; n = 147 for *Ppa-tbh-1* +Histamine, n = 79 for *Ppa-tdc-1*::HisCl -Histamine, n = 138 *Ppa-tdc-1*::HisCl +Histamine. (**c**) Average transition rates between behavioural states for *tbh-1* + 10 mM Histamine, *tdc-1p*::HisCl without histamine and *tdc-1p*::HisCl + 10 mM histamine silencing conditions. The number in circles indicates the average state duration as in (b), and the arrow size indicates the transition rate normalized to outgoing transitions.

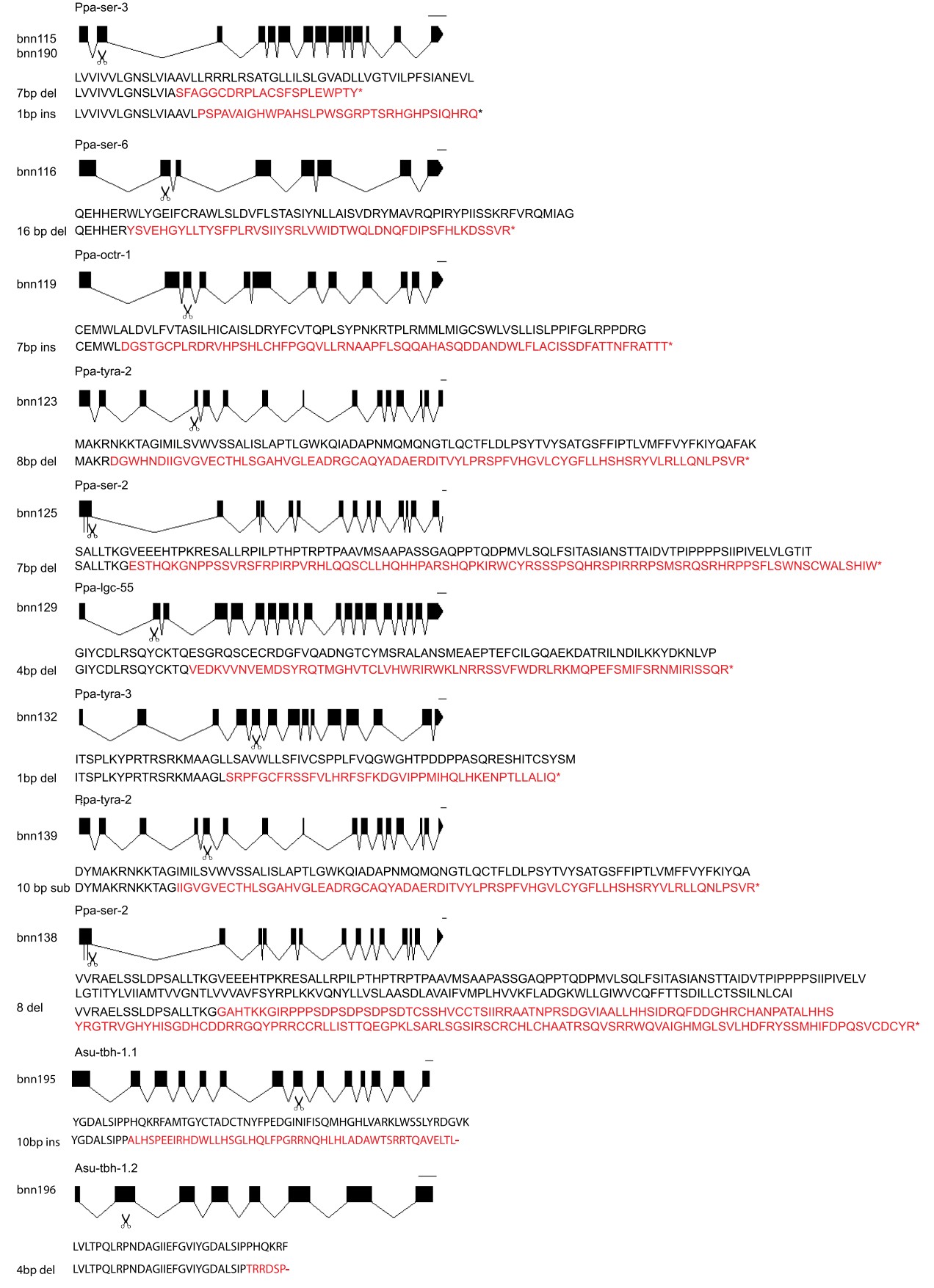

**Extended Data Fig. 9 | CRISPR/Cas9 mutants generated.** Gene structures and protein sequence alignments of all mutants generated by CRIPSR/Cas9 in this study. Schematics generated using wormweb.org exon-intron graphic maker. Scale bar = 100 bp.

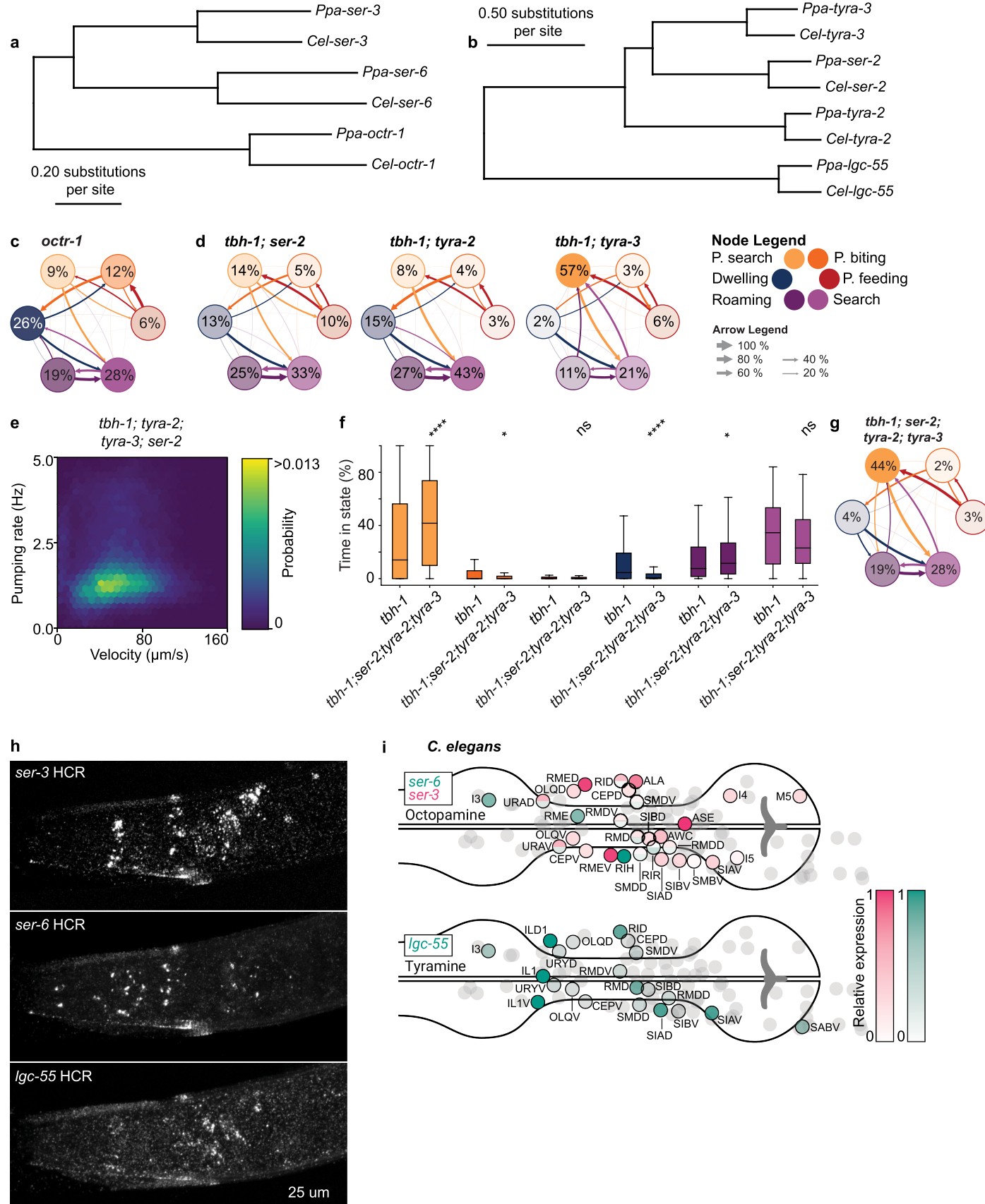

**Extended Data Fig. 10** | See next page for caption.

**Extended Data Fig. 10 | Behavioural state prediction for *tph-1, cat-2*, and *tbh-1; tdc-1* and sensory anatomy.** (**a**) Phylogenetic analysis of octopamine receptor genes from *P. pacificus* and *C. elegans*. The tree supports a single orthologous relationship between species. (**b**) Phylogenetic analysis of tyramine receptor genes from *P. pacificus* and *C. elegans*. The tree supports a single orthologous relationship between species. (**c**) Average transition rates between behavioural states for the octopamine receptor *octr-1* and (**d**) the tyramine receptor genes *tbh-1; ser-2, tbh-1; tyra-2*, and *tbh-1; tyra-3*. The number in circles indicates the average state duration and the arrow size indicates the transition rate normalized to outgoing transitions. (**e**) Probability density map of velocity and pumping rate for the quadruple mutant *tbh-1; ser-2; tyra-2; tyra-3*.

(**f**) Relative time in each behavioural state for WT compared to *tbh-1; ser-2; tyra-2; tyra-3* quadruple mutant. Statistics were performed with Mann-Whitney-U-test using Bonferroni correction for multiple tests against *Ppa-tbh-1*. *p < 0.05, **p < 0.01, ***p < 0.001, ****p < 0.0001; n = 197 for *Ppa-tbh-1*, n = 181 for *Ppa-tbh-1-ser-2-tyra-2-tyra-3*. (**g**) Average transition rates between behavioural states for the *tbh-1; ser-2; tyra-2; tyra-3* quadruple mutant. (**h**) mRNA expression of *ser-3, ser-6* and *lgc-55* in the same animal using HCR probes. Anterior is left. (**i**) Expression pattern of the *C. elegans* receptors *ser-3* (magenta) and *ser-6* (green), and *lgc-55* (green) in the animal head, based on data from the CeNGEN single-cell transcriptome analysis. Gray dots represent neurons that don't show expression.

# Reporting Summary

## Statistics

For all statistical analyses, confirm that the following items are present in the figure legend, table legend, main text, or Methods section.

| n/a | Confirmed | |
|---|---|---|
| ☐ | ☒ | The exact sample size (*n*) for each experimental group/condition, given as a discrete number and unit of measurement |
| ☐ | ☒ | A statement on whether measurements were taken from distinct samples or whether the same sample was measured repeatedly |
| ☐ | ☒ | The statistical test(s) used AND whether they are one- or two-sided *Only common tests should be described solely by name; describe more complex techniques in the Methods section.* |
| ☒ | ☐ | A description of all covariates tested |
| ☐ | ☒ | A description of any assumptions or corrections, such as tests of normality and adjustment for multiple comparisons |
| ☒ | ☐ | A full description of the statistical parameters including central tendency (e.g. means) or other basic estimates (e.g. regression coefficient) AND variation (e.g. standard deviation) or associated estimates of uncertainty (e.g. confidence intervals) |
| ☐ | ☒ | For null hypothesis testing, the test statistic (e.g. *F*, *t*, *r*) with confidence intervals, effect sizes, degrees of freedom and *P* value noted *Give P values as exact values whenever suitable.* |
| ☒ | ☐ | For Bayesian analysis, information on the choice of priors and Markov chain Monte Carlo settings |
| ☒ | ☐ | For hierarchical and complex designs, identification of the appropriate level for tests and full reporting of outcomes |
| ☒ | ☐ | Estimates of effect sizes (e.g. Cohen's *d*, Pearson's *r*), indicating how they were calculated |

*Our web collection on statistics for biologists contains articles on many of the points above.*

## Software and code

Policy information about availability of computer code

| Data collection | Data was collected with commercially available microscopes or custom imaging setups as described in detail in the methods section. |
|---|---|
| Data analysis | The raw imaging data was analyzed using the published, open source package PharaGlow v0.9.6 (https://doi.org/10.7554/eLife.77252). The code used for behavioral state detection is described in the manuscript and available under a GPL 3.0 license: https://github.com/scholz-lab/PpaPred. PharaGlow and PpaPred require Python 3.8. |

For manuscripts utilizing custom algorithms or software that are central to the research but not yet described in published literature, software must be made available to editors and reviewers. We strongly encourage code deposition in a community repository (e.g. GitHub). See the Nature Portfolio guidelines for submitting code & software for further information.

## Data

Policy information about availability of data

All manuscripts must include a data availability statement. This statement should provide the following information, where applicable:
- Accession codes, unique identifiers, or web links for publicly available datasets
- A description of any restrictions on data availability
- For clinical datasets or third party data, please ensure that the statement adheres to our policy

Data availability
All data are available in the main text or the supplementary, or tracking data on OSF at https://osf.io/cua87. Raw data from behavioral tracking will be made

available via file transfer upon reasonable request, as these data are > 200 GB.

## Research involving human participants, their data, or biological material

Policy information about studies with human participants or human data. See also policy information about sex, gender (identity/presentation), and sexual orientation and race, ethnicity and racism.

| Reporting on sex and gender | NA |
| --- | --- |
| Reporting on race, ethnicity, or other socially relevant groupings | NA |
| Population characteristics | NA |
| Recruitment | NA |
| Ethics oversight | NA |

Note that full information on the approval of the study protocol must also be provided in the manuscript.

# Field-specific reporting

Please select the one below that is the best fit for your research. If you are not sure, read the appropriate sections before making your selection.

☒ Life sciences    ☐ Behavioural & social sciences    ☐ Ecological, evolutionary & environmental sciences

For a reference copy of the document with all sections, see nature.com/documents/nr-reporting-summary-flat.pdf

# Life sciences study design

All studies must disclose on these points even when the disclosure is negative.

| Sample size | Sample sizes were set to allow sufficient sampling of the joint distribution of velocity, and feeding rates, the major features used in the analysis. The estimate was based on the methods paper describing the analysis for C. elegans in https://doi.org/10.7554/eLife.77252 |
| --- | --- |
| Data exclusions | Data exclusions followed the procedure described in https://doi.org/10.7554/eLife.77252. In brief, animals that were sampled for less than a minimum period were automatically excluded without regard to genotype or condition. |
| Replication | Experiments were conducted at least in triplicate to ensure reproducibility. |
| Randomization | Behavioral assays were performed in random order to avoid batch effects. |
| Blinding | Strong phenotypes are readily apparent to the experimenter and therefore blinding is impractical. Analysis was performed without regard to genotype or condition, thus ensuring blind analysis. |

# Reporting for specific materials, systems and methods

We require information from authors about some types of materials, experimental systems and methods used in many studies. Here, indicate whether each material, system or method listed is relevant to your study. If you are not sure if a list item applies to your research, read the appropriate section before selecting a response.

## Materials & experimental systems

| n/a | Involved in the study |
| --- | --- |
| ☒ | ☐ Antibodies |
| ☒ | ☐ Eukaryotic cell lines |
| ☒ | ☐ Palaeontology and archaeology |
| ☐ | ☒ Animals and other organisms |
| ☒ | ☐ Clinical data |
| ☒ | ☐ Dual use research of concern |
| ☒ | ☐ Plants |

## Methods

| n/a | Involved in the study |
| --- | --- |
| ☒ | ☐ ChIP-seq |
| ☒ | ☐ Flow cytometry |
| ☒ | ☐ MRI-based neuroimaging |

# Animals and other research organisms

Policy information about [studies involving animals](); [ARRIVE guidelines]() recommended for reporting animal research, and [Sex and Gender in Research]()

| | |
|---|---|
| Laboratory animals | Nematodes - C. elegans, P. pacificus, A. suhausi (all strains are detailed in a strain table in the manuscript). All strains are hermaphroitic and were tested at the young adult stage. |
| Wild animals | no wild animals were used in this study. |
| Reporting on sex | The study organism is hermaphroditic. |
| Field-collected samples | No field collected samples were used in the study. |
| Ethics oversight | No ethics approval was required for any experiments (invertebrate research). |

Note that full information on the approval of the study protocol must also be provided in the manuscript.

# Plants

| | |
|---|---|
| Seed stocks | n.a. |
| Novel plant genotypes | n.a. |
| Authentication | n.a. |

