## [Peer Review File · Nature]

Predatory aggression evolved through adaptations to noradrenergic circuits

Corresponding Author: Dr James Lightfoot

Version 0:

Reviewer comments:

Referee #1

(Remarks to the Author)

In this manuscript, Eren, Boger et al. provide a characterization of the neuromodulatory systems that control aggressive behaviors in *P. pacificus*, comparing the circuit organization to that of *C. elegans*. First, they develop a machine learning classifier to quantify the behavioral states of *P. pacificus*. This identifies states known to occur in *C. elegans*, plus additional states associated with predation that are specific to *P. pacificus*. The authors further show that behavioral state occupancy differs dramatically if animals are feeding on bacteria versus feeding on other nematodes. The authors then examine the functional contributions of neuromodulators to these states. This reveals that octopamine induces states associated with predation, whereas tyramine inhibits those states. The authors show that these neuromodulators, as well as their receptors, are expressed in groups of cells that differ from those where they are found in *C. elegans*, suggesting potential adaptations that may have allowed *P. pacificus* to evolve its distinct feeding behaviors.

In general, this is a very exciting manuscript with a high level of novelty. The molecular and circuit adaptations that give rise to distinct behavioral states over evolutionary time are poorly understood and the authors have an excellent case study here. Nevertheless, I do have some suggestions, mostly aimed at firming up the data/analyses to strengthen support for the conclusions. Provided that the evidence is tightened up, I'd be supportive of publication in Nature.

Major concerns:

1. The machine learning approach is interesting, but there were a few questions. First, the authors perform feature extraction and clustering on a dataset of 106 animals. This clustering reveals six states, which form the basis of the entire paper. It would be important to show that a totally independent dataset reveals the same 6 clusters using the same approach – i.e. that this is a robust observation. The split of training/testing for the classifier was fine, but I am concerned that the basic clustering (a premise of even building the machine learning classifier) was not shown to be robust across datasets/animals. (2 additional minor questions: Were the 106 animals inclusive of animals on bacteria and on nematode diets? Could the authors validate that the predatory biting/feeding states occur specifically when the animal is in contact with larvae as an additional means of validating the states?)
2. I am somewhat concerned about the robustness of the *tbh-1*/octopamine phenotype. There is a good mutant phenotype in Fig. 3 that I initially found persuasive. However, this is not recapitulated in the independent data in Fig. 4b (as far as I can tell?). In addition, exogenous octopamine did not really have much an effect on these states. The genetic interaction with *tdc-1* is interesting, albeit unusual. I'd suggest firming up this basic *tbh-1* mutant phenotype and, if exogenous octopamine does not work to change these states, obtaining an additional line of evidence to support this hypothesized function for octopamine. Even a second, independent allele of *tbh-1* would help.
3. Regarding the neurons labeled in Fig. 3f. This image is a bit poor resolution and unlabeled, so I'd suggest improving. In addition, I'd suggest using a different genetic marker (potentially a transcription factor that marks cell identity) for RIM and RIC to really validate that these cells are RIM and RIC, as the authors believe. In addition, was the anterior projection of RIM noted in the recent EM of *P. pacificus*? If so, that would provide further support that that cell is indeed RIM. I'd also be interested to see RIM- or RIC-specific rescues of *tbh-1* and *tdc-1*, though I understand that this would depend on availability

of cell-specific promoters.

4. The authors raise an interesting hypothesis about IL and OL neurons: these neurons may be involved in contact with nematode prey and modulated by tyramine/octopamine to alter how animals respond to those incoming prey-related sensory cues. However, there really needs to be further experiments here. I'd suggest both (i) silencing these neuron classes and examining the behavioral impact; and (ii) performing genetic rescue of the receptors in these cells. This would provide the basic evidence necessary to support the authors' hypothesis.

5. I'm unclear how well conserved the amino acid sequences of the octopamine and tyramine receptors are. If they are not identical, are the authors sure that these receptors definitely respond to the same ligands? For example, some of these receptors (e.g. ser-3) were originally misnamed as presumed serotonin receptors based on homology, but subsequently shown to actually respond to different amines. The authors should comment on this and, if the sequences are divergent, provide evidence (genetic or biochemical) that the main receptor(s) of interest are indeed selectively activated by tyramine or octopamine.

Minor comments:

1. Please add WT to Extended data fig. 5a.

2. Please add WT to Fig. 4e

3. There is a typo (missing character?) on line 342 that was confusing.

4. I don't usually care about semantics, but I did find myself wondering whether aggression was really the correct term here. Why not consummatory behavior? Tbh-1 (at least in Fig. 3) has reduced biting, but also reduced dwelling. Dwelling is a non-aggressive consummatory behavior. Perhaps the authors could weigh alternatives and justify their terminology choice in the Results section.

(Remarks on code availability)

Referee #2

(Remarks to the Author)

The precise genetic, molecular, and neural mechanisms shaping species differences in behavioral variation are poorly understood, especially in the context of the evolution of novel behaviors. The evolution of intraguild predation behavior in *P. pacificus*, where individuals kill and prey on different species to both reduce competition and fulfill nutritional needs, provides an excellent model system. Here, the authors have developed genetic reagent that enables the application of a machine learning-based approach to define the various behavioral states associated with predatory behavior, providing an unbiased, high-throughput assay to examine the roles of noradrenergic pathways. Specifically, based on the mutant phenotypes of neuromodulators and their receptors, the authors provide evidence that the octopamine pathway promotes predatory biting, while the tyramine pathway acts antagonistically to induce non-predatory states. The authors further identified a few putative species differences, including an additional pair of octopaminergic neurons with neurite projecting to the mouth region associated with feeding. This neurite is likely absent in *C. elegans*. The authors also identified that this neuron potentially expresses an octopamine receptor, raising the possibility of a self-reinforcing motif that may contribute to a persistent predatory state. Finally, the authors identified possible species differences in the expression pattern of octopamine receptor neurons in *P. pacificus*, including neurons that project to head sensory neurons extending to the nose. Together, the authors suggest that noradrenergic circuits balance aggressive behavioral states in nematodes, and evolutionary adaptations in the octopamine/tyramine circuits facilitate the evolution of predatory behaviors, potentially by promoting a persistent aggressive state and state-dependent gating of sensory neurons. By addressing the neural mechanisms for a well-known behavioral innovation and the neuromodulatory mechanisms regulating such a complex behavior, the work is of potential interests to a broad audience.

Major comments

The role of noradrenergic circuits in predatory behaviors in *P. pacificus* has been convincingly established in this study. Nevertheless, the main conclusion of this manuscript—that evolutionary adaptations in octopaminergic circuits have facilitated the emergence of aggressive behavioral states associated with complex predatory traits—has yet to be fully established. To convincingly support this conclusion, the following points need to be addressed: (1) clearly define the species differences; (2) examine the phenotypic outcomes of key species differences; and (3) demonstrate that these species differences represent predatory-specific adaptations. Additionally, caution is needed when interpreting whether the phenotypes observed in predatory behaviors (such as biting) can be directly considered aggressive states. Below, I will discuss each of these points in more detail.

First, the authors have identified several notable species differences. The most important is the gain of a pair of octopaminergic neurons. Specifically, the putative RIM is both tyraminerpic and octopaminergic in *P. pacificus*, while it is only tyraminerpic in *C. elegans*. This conclusion is based on a transgenic reporter assay, which has its limitations (e.g., genomic location effects). Given the importance of this finding for the manuscript's overall conclusion, further validation is required, such as a CRISPR knock-in of the reporter gene at its endogenous location. If HDR-directed CRISPR is not established in *P. pacificus*, can the authors consider alternative methods like *in situ* hybridization. Furthermore, the putative RIM neuron in *P. pacificus* exhibits a neurite extending anteriorly toward the mouth, a feature absent in *C. elegans*. Given the reported differences in both the noradrenergic identity and morphology of RIM between the species, the homology of RIM should be strengthened, such as by examining characteristic branching patterns of RIM, if any. A recently published preprint comparing the EM connectomes of the two species (Cook et al. 2024) noted the conservation of characteristic branching

patterns in RIM but did not mention any species-specific pattern. Additionally, based on the results in this preprint, *P. pacificus*'s RIM did not seem to exhibit major *pacificus*-specific postsynaptic connections. While these EM data are not definitive, the potential inconsistency with the interpretation from the current study calls for caution of inferring homology. Strong evidence for homology is a necessary foundation for the authors' subsequent conclusions about rewiring and morphological divergence. Finally, the authors identified species differences in the expression patterns of octopamine receptors, with RIM potentially acting as a *P. pacificus*-specific site for a positive auto-receptor motif. The possibility that RIM functions as such a motif to promote a persistent predatory state is exciting but could benefit from simple validation (e.g., *in situ* hybridization to confirm that the putative RIM is also *ser-6* positive). Similarly, considering the different methods used for species comparison (reporter assays in *P. pacificus* and scRNA-seq in *C. elegans*), important candidate species differences (such as IL1 and IL2) should be further validated applying the same method.

Second, despite identifying many potentially relevant species differences, the phenotypic outcomes of these differences were not directly examined. While I understand the technical challenges, functional validation is critical to make the stated inferences. Particularly, *C. elegans* and *P. pacificus* are highly divergent species (100-300 million years ago), and many changes may have occurred. For example, the EM connectome has identified widespread differences between the two species in cell number, projection patterns, cell body positions, and synaptic connectivity, but not all of these differences are expected to be relevant for the predatory behavior. Furthermore, not all changes are necessarily functional—developmental and circuit drift are common phenomena in evolution (for example, see the connectome paper for examples of developmental drift). At a minimum, the authors should attempt to address the functional consequences of key species differences, such as using RIM-specific RNAi of *tbh-1* or *ser-6*. I understand that these experiments may or may not be readily possible, but without such support, the major conclusion of the paper lacks direct support.

Third, do species differences observed readily represent *P. pacificus*/lineage-specific adaptations for its unique predatory behavior? I am afraid not. The notes species differences (e.g., the putative RIM being octopaminergic in *P. pacificus* but not in *C. elegans*) could be either a gain in *P. pacificus* or a loss in *C. elegans*. Phylogenetic support (e.g., outgroup species of *P. pacificus* without predatory feeding does not exhibit the additional pair of octopaminergic neurons) is needed to distinguish the two possibilities, especially given the long evolutionary divergence time.

Fourth, the authors switched from using the term 'predatory behavior' in the Results section to 'aggressive behavior' in the Discussion (Abstract as well) without providing a clear rationale for this interchange. This should be handled with great caution. First, although intraguild predation behavior involves aggression with the goal of killing the competitor, it also results in obtaining nutritional resources. Aggression has typically been distinguished from predation due to its nutritional goals (Quach and Chalasani et al., 2020). Predatory biting can be driven by both goals, and this distinction cannot be made from the current assay. Further, while a reasonable hypothesis, there is no direct evidence supporting that the octopamine/tyramine circuits encode the aggressive state. To support the claim that octopaminergic neurons encode aggression (or that octopamine/tyramine neurons gate the aggressive state, as stated in the title of Figure 4), as the minimal, evidence such as neuron activation leading to persistent biting but not feeding is needed. Additionally, since octopamine injection did not increase biting (line 161, ED Fig. 6), interchanging the two terms is particularly concerning.

Minor comments

Line 161: Can the authors speculate why octopamine injections didn't increase aggressive behavior?

Line 162: The statement that "evolutionary divergence in the function of these neuromodulators" is potentially problematic, especially when "function" is not clearly defined. My opinion is that the term "evolutionary divergence" is typically not used in such context. For example, dopamine may promote different behaviors in a species-specific manner, but the function of dopamine as a rewarding signal remains the same, and the dopaminergic circuit also does not have to be functionally divergent. In the current study, the more direct interpretation is that "these neuromodulators regulate this novel behavior."

Error bars are not defined in figure legends throughout the main and ED figures.

Figure 3d: Dopamine-deficient mutants show decreased 'predatory search' and 'predatory feeding', but not 'predatory biting.' Based on this observation, the authors predicted that dopamine may be required for initiating feeding after a kill. This does not explain why 'predatory search' also decreases. It is also unclear why decreased 'predatory search' did not lead to decreased biting. Would this imply that the probability of biting following a search increases?

Figures 4d and 3d: The phenotypes of *tbh-1* mutants appear quite different between the two figures? Would this difference be large enough to impact the robustness of the conclusions? For example, if the *tbh-1* data in figure 4 were replaced with those from figure 3, would the conclusions of all the comparisons remain the same?

Figure 4e: The role of *ppa-lgc-55* is the strongest among the four receptors; however, other tyramine receptors may also play a role. Not fully restoring the phenotype to the wild-type level does not necessarily mean that they don't play a role. The authors stated that double mutants of *tbh-1* and individual tyramine receptors were tested to see if any tyramine receptor could rescue the *tbh-1* phenotype in reducing aggressive biting. If so, why were all the comparisons made to *tdc-1* (to test for full rescue) instead of *tbh-1* (to test for any rescue)?

Line 225: The authors suggest that the auto-receptor motif may contribute to persistent and robust predatory episodes. If so, would the measurement of "state duration" (in addition to "state time") also be directly relevant? However, for the two feeding conditions in wild-type animals, the "state duration" remains largely unaltered (ED Fig. 4).

Are the functions of the sensory neurons that express octopamine receptors known? If relevant, this may provide support for the hypothesis of state-dependent gating of sensory neurons in regulating aggressive behavior. If not, the tone for this interpretation shall be greatly weakened at least.

Line 399: Method: "predictions with a probability of less than 50% are set to 'None'." Can the authors provide a rationale for this threshold? How is the probability threshold for calling a prediction decided? Is it based on balancing precision and recall? I could not find relevant information. Please include it if it is indeed missing.

Figures 4g and 4h could benefit from improved readability by (1) labeling neurons of central interest in this study (RIM, IL1, IL2) in *P. pacificus* and (2) labeling where the head is.

Line 120: "Thus, we observe greatly expanded state complexity in *P. pacificus*, including the presence of aggressive episodes, which have not been previously reported in other nematodes, including *C. elegans*." I don't quite understand this sentence since *C. elegans* does not exhibit these behaviors. For me, it simply means that "the pipeline successfully captures the various states associated with predatory behaviors in *P. pacificus*."

Ground truth for machine learning: How do the authors assign states, and what is the source of the ground truth? For example, is the assignment of 'predatory biting' blind to information on the presence/absence of *C. elegans* larvae? In other words, would any clips from the bacterial lawn be defined as 'predatory biting' and 'predatory feeding'? If so, what percentage of these clips come from the *C. elegans* larvae condition compared to other conditions?

Could the authors provide a panel comparing RIM cell body location and morphology between species side by side and noting the *P. pacificus*-specific phalangeal-projecting neurite?

Figure 1f: I guess this panel represents the average of all states? If so, please note it in the figure legend. Also remove the redundant data from Extended Data Figure 3.

Line 288: "two pairs of neurons expressing both neurotransmitters with novel features"— RIM is the only one with novel features?

(Remarks on code availability)

Referee #3

(Remarks to the Author)

This study was a total pleasure to read. They ask penetrating questions, use powerful tools with precision and thoroughness, expand our understanding of noradrenergic pathway from one nematode species to another, and discover interlocking patterns involving evolutionary conservation and divergence at all levels from signaling molecules to receptors to cells to behaviors.

Pristionchus pacificus has a fraction of the tools that are available in *C. elegans*, and yet they manage (with CRISPR) to effectively dissect its noradrenergic circuits. I marvel at the intelligence of their new behavioral and segmentation assays. I marvel at the clarity and relentless logic of their prose.

The authors study the neural and molecular basis of *Pristionchus* predation. They express RFP in the pharyngeal muscles (unlike in *C. elegans*, the fluorescence is only visible in the corpus) and record animals in varying environments and under various genetic perturbations. The paper leverages a previously built pipeline called PharaGlow (Bonnard et al. 2022), which allows the authors to extract features from the recordings of the fluorescently labeled pharyngeal muscles. Via unsupervised clustering (UMAP followed by hierarchical clustering), the authors identify six distinct behavioral states, corresponding to roaming, dwelling, searching, predatory feeding, predatory biting and predatory searching. An expert annotator then creates ground truth for recordings and a classifier is trained to label recordings into one or none of these six states. The authors use plots such as the probability density of pumping rates and velocities, the transition probabilities between states and the percentage time an animal spends in a certain state to study how environment and neuromodulators affect predatory behavior.

The authors observe that *Pristionchus* predation related behaviors almost exclusively occur on larvae. The authors then ask if neuromodulators play an interesting role in this transition, and screen four mutants based on orthologues in: *\emph{tph-1}* (serotonin), *\emph{cat-2}* (dopamine), *\emph{tbh-1}* (octopamine) and *\emph{tdc-1}* (tyramine, and by proxy octopamine). Consistent with previous literature, they find similar implications of serotonin in predatory behavior.

The authors also find that *\emph{cat-2}* mutants have a reduction in predatory feeding and predatory search, but not in biting behavior, implying that dopamine could be a food reward signal the animal receives after a successful kill. Interestingly, the authors also show the role of octopamine in the induction of a prolonged predatory state, and tyramine in the induction of a docile state. They identify neurons RIM and RIC as both tyramineric and octopaminergic, and observe an extra neurite in RIM that makes it distinct from the *C. elegans* RIM.

The authors then study receptors of octopamine and tyramine by making mutants of orthologues of known *C. elegans* receptors.

They observe that cells expressing *ser-3* and *ser-6* octopamine receptors are important in predation via the octopamine signaling pathway. Cells expressing *lgc-55*, a tyramine receptor, are important in the docile state induced by the tyramine signaling pathway. Using reporter lines, they observe that these receptors are expressed in neck muscles and a few head neurons. Interestingly, octopamine receptors can be found in IL1 (mechanosensory), IL2 and RIM (expression in RIM implies some feedback). Tyramine receptors seem to be expressed in a distinct set of head neurons, one of which seems to be OL.

All I really had to do in this Review is document their beautifully done work. In the end, my only real question was how they might prove that the bistable state-switch (octopamine release stimulating further octopamine release), actually works this way. It suggests a truly elegant cell-autonomous mechanism for sustaining an octopamine-dependent behavioral state. The hypothesis is reasonable, and I couldn't think of a reasonable experiment that would cleanly test the hypothesis. The bistable state-switch model is adequately presented as food for thought.

In closing, this is a seriously well-done study and well-written paper. Congratulations!

(Remarks on code availability)

Version 1:

Reviewer comments:

Referee #1

(Remarks to the Author)

The authors provide a revised manuscript that has some very nice additions. For the most part, I am happy with the revisions. I still believe the study should ultimately be published in *Nature*. However, I was surprised to see that two important questions were still unresolved in the revised manuscript, even though they were brought up by two reviewers including myself:

1. Which neurons make tyramine and octopamine in *P. pacificus* to regulate behavioral states? In the initial submission, there were claims that RIM produced both tyramine and octopamine and had an anterior projection. These very surprising findings were removed from the study. I'm guessing that the authors no longer believe those results, which is fine. What is left in the revised manuscript is HCR labeling of *tdc-1* and *tbh-1*. While the number of labeled neurons now matches the *C. elegans* results, the labeling of neuronal identity from these images is at best a guess (nobody could confidently label the identities of these cells from those pictures). In addition, the authors did not provide requested genetic rescues of *tdc-1* or *tbh-1* in RIM or RIC. Unfortunately, this leaves a pretty central part of the model unresolved. Is RIC the functional site of octopamine production relevant to these behaviors? Is RIM the functional site of tyramine production? The final model cartoon in the paper (Fig.6i) depicts it as if this was shown to be the case. I hate to add extra work, as I appreciate that the authors have already done a lot, but I do think the bare minimum to resolve this is important. There are abundant reporter gene studies for other genes and transgenic lines (for *HisCl* expression, etc) elsewhere in the study. I'd suggest the author confirm that RIM and RIC are the relevant neurons that produce these neuromodulators (via *tdc-1* and *tbh-1* reporter genes or HCR labeling overlapping with genetic markers for those 2 cells). In addition, they should silence RIM and RIC each on their own (or perform cell-specific rescues of *tdc-1* and *tbh-1*, respectively) and examine the consequences for behavioral states.

2. The IL2::*HisCl* experiment is nice. But it is still not clear that octopamine receptor expression in IL2s has any functional importance. Given that the authors have a good IL2-specific promoter, could the authors perform a genetic rescue of key octopamine receptors and see if this rescues behavior? I had also suggested this experiment in my previous review.

Again, I hate to ask for more work. But the model cartoon in Fig. 6i that is meant to summarize the paper indicates that RIM/RIC release tyramine/octopamine to signal to IL2/OL neurons via specific receptors SER-3/SER-6/LGC-55. Currently, this simply isn't supported by data and I don't see how I can suggest publication with that being the case. I am happy to recommend publication in *Nature* once these last questions are resolved.

(Remarks on code availability)

Referee #2

(Remarks to the Author)

The authors have thoroughly and rigorously addressed all of my earlier concerns. This is a beautiful piece of work that offers important insights into both the neuromodulatory mechanisms underlying behavioral state regulation and their role in driving behavioral innovation. I have no remaining reservations and strongly support its publication.

(Remarks on code availability)

Version 2:

Reviewer comments:

Referee #1

(Remarks to the Author)

I am happy to now recommend publication. I appreciate the tdc-1::HisCl experiment and best efforts with ser-3. It would be worthwhile to describe detailed methods for transgenesis in the Methods section of this study, as it sounds like the authors are leading experts and some challenges remain. Detailed methods could help others further develop this exciting model system.

Congratulations on a landmark study.

(Remarks on code availability)

[REDACTED]

We greatly appreciate the positive review of our manuscript 'Predatory aggression evolved through adaptations to noradrenergic circuits' and thank the reviewers for their constructive comments. The referees found it a 'beautifully done work' with 'a high level of novelty', with one reviewer specifically pointing out the depth of the work 'involving evolutionary conservation and divergence at all levels from signaling molecules to receptors to cells to behaviors'.

The reviewers also formulated some concerns, which we have managed to fully address in our substantially revised manuscript. In brief, we have added further evidence supporting the robustness of our model, including an entirely new figure demonstrating the validity of the model predictions by visualizing predator and prey simultaneously, thus confirming the predictions of predatory aggressive states. We also validated our expression data for both neurotransmitters and receptors using an additional independent technique, and have substantially increased the support for our claims regarding the evolution of predatory aggression, as well as added further supporting evidence that these states indeed reflect aggressive, rather than nutrient-driven behavior.

We hope that with these substantial additions our manuscript is now suitable for publication in *Nature*.

Best regards,
Monika Scholz and J.W Lightfoot

You find the referee's comments presented below in black with our point-by-point response in red.

Referees' comments:

Referee #1 (Remarks to the Author):

In this manuscript, Eren, Boger et al. provide a characterization of the neuromodulatory systems that control aggressive behaviors in *P. pacificus*, comparing the circuit organization to that of *C. elegans*. First, they develop a machine learning classifier to quantify the behavioral states of *P. pacificus*. This identifies states known to occur in *C. elegans*, plus additional states associated with predation that are specific to *P. pacificus*. The authors further show that behavioral state occupancy differs dramatically if animals are feeding on bacteria versus feeding on other nematodes. The authors then examine the functional contributions of neuromodulators to these states. This reveals that octopamine induces states associated with predation, whereas tyramine inhibits those states. The authors show that these neuromodulators, as well as their receptors, are expressed in groups of cells that differ from those where they are found in *C. elegans*, suggesting potential adaptations that may have allowed *P. pacificus* to evolve its distinct feeding behaviors.

In general, this is a very exciting manuscript with a high level of novelty. The molecular and circuit adaptations that give rise to distinct behavioral states over evolutionary time are poorly understood and the authors have an excellent case study here. Nevertheless, I do have some suggestions, mostly aimed at firming up the data/analyses to strengthen support for the conclusions. Provided that the evidence is tightened up, I'd be supportive of publication in *Nature*.

We thank the reviewer for their insightful and very positive evaluation of our work. Please find below our response to the reviewer's comments which we believe fully address their concerns.

Major concerns:

1. The machine learning approach is interesting, but there were a few questions. First, the authors perform feature extraction and clustering on a dataset of 106 animals. This clustering reveals six

states, which form the basis of the entire paper. It would be important to show that a totally independent dataset reveals the same 6 clusters using the same approach – i.e. that this is a robust observation. The split of training/testing for the classifier was fine, but I am concerned that the basic clustering (a premise of even building the machine learning classifier) was not shown to be robust across datasets/animals. (2 additional minor questions: Were the 106 animals inclusive of animals on bacteria and on nematode diets? Could the authors validate that the predatory biting/feeding states occur specifically when the animal is in contact with larvae as an additional means of validating the states?)

We appreciate the reviewer's suggestion and helpful comments. To strengthen the support for our behavioral pipeline, we have now analysed an entirely independent dataset using the exact same parameters, thus confirming robustness of the observed behavioral states. The original dataset, as well as this new validation dataset contain a mix of animals in both predatory and bacterial environments, which ensures that the clustering algorithm has access to all possible states. We also ensured that the validation dataset had roughly similar numbers for each condition to make the comparison fair. Given the robustness across these repeated datasets, we are convinced that these states indeed reflect valid behavioral states.

In addition to further validate the predatory states and answer the reviewer's question if biting is indeed specific to contact with larval prey, we constructed and utilized a novel dual-color fluorescence tracking microscope to visualize predator-prey interactions directly. By tracking individual predators as they interacted with labelled prey, we could observe that when our model predicted aggressive predatory events this coincided with a nose-contact between predator and prey. In addition, we also validated that ingestion co-occurred with predicted feeding, therefore further validating our state annotations. These data are now presented as a new Figure 3. Furthermore, a manuscript describing the design and software underlying this specialized microscope are available on BioRxiv (doi: 10.1101/2025.04.01.646316). We are grateful to the reviewer for giving us the impetus to add this additional technique to the manuscript.

2. I am somewhat concerned about the robustness of the *tbh-1*/octopamine phenotype. There is a good mutant phenotype in Fig. 3 that I initially found persuasive. However, this is not recapitulated in the independent data in Fig. 4b (as far as I can tell?). In addition, exogenous octopamine did not really have much an effect on these states. The genetic interaction with *tdc-1* is interesting, albeit unusual. I'd suggest firming up this basic *tbh-1* mutant phenotype and, if exogenous octopamine does not work to change these states, obtaining an additional line of evidence to support this hypothesized function for octopamine. Even a second, independent allele of *tbh-1* would help.

We appreciate this helpful comment and apologize for not being clearer with the difference in statistical comparison used for Fig. 3 and Fig. 4b which has caused this confusion (now Figure 4 and Figure 5). For Fig. 3 we conducted our statistical comparison between wildtype and the mutants which showed a clearly significant decrease in predatory aggressive states in the *tbh-1* mutant. For Figure 4b we compared all of the receptors against the *tbh-1* mutant to see if they phenocopied this mutant and were in the same pathway. Each dataset shown in these figures is indeed an independent dataset and they fully corroborate and agree with one another. We have highlighted this change in comparison in the figure caption for additional clarification.

We have now further explored the exogenous octopamine experiments to include additional mutants. This includes the addition of octopamine onto the *tbh-1* and *tdc-1* mutants. The addition of octopamine into the null mutant background rescues the killing phenotype reinforcing its role but does not increase it further, similar to the addition in the wildtype background. Therefore, we interpret that the maximum predatory aggressive potential has been reached and the system is saturated. This is now included in Extended data figure 6.

We agree that the genetic interaction between *tbh-1* and *tdc-1* is very interesting and in order to clarify this further we have now included a second independent allele of both *tbh-1* and *tdc-1* which both phenocopy our original alleles. In summary, these data robustly confirm our previous phenotypes for these two neurotransmitters.

3. Regarding the neurons labeled in Fig. 3f. This image is a bit poor resolution and unlabeled, so I'd suggest improving. In addition, I'd suggest using a different genetic marker (potentially a transcription factor that marks cell identity) for RIM and RIC to really validate that these cells are RIM and RIC, as the authors believe. In addition, was the anterior projection of RIM noted in the recent EM of *P. pacificus*? If so, that would provide further support that that cell is indeed RIM. I'd also be interested to see RIM- or RIC-specific rescues of *tbh-1* and *tdc-1*, though I understand that this would depend on availability of cell-specific promoters.

We agree with the reviewer that establishing cell identity across large evolutionary timescales is a complex procedure that has not yet been fully established. To further determine the identity of our *tbh-1* and *tdc-1* expressing cells we attempted several methods. This included using Cel antibodies raised against *tbh-1* and *tdc-1*, generating additional transcriptional reporters against common molecular markers for these neurons used in *C. elegans* research (specifically *Ppa-cex-1*) and Hybridization chain reaction to detect transcripts directly (HCR). A recent review by Oliver Hobert (<https://www.nature.com/articles/s41583-021-00497-x>) raised the issue of establishing cell identity and discusses the (dis-)advantages of all these methods. While transcriptional reporters are often the de facto state-of-the-art even in *C. elegans*, these may not fully capture the endogenous expression patterns. In contrast, antibodies are sometimes non-specific or don't bind well enough, while mRNA-based methods are capturing only a snapshot of cellular expression. Unfortunately, as *P. pacificus* does not have all possible techniques established at this point, we attempted to port relevant approaches from *C. elegans*. However, only HCR was successful in marking *tbh-1* and *tdc-1* expressing cells but only show the cell soma and by their design only detect current transcripts. Using this method, we detected a single pair of neurons expressing *tbh-1* and two pairs of neurons expressing *tdc-1*. This is similar to observations in *C. elegans*. We have adjusted our figure in line with the HCR evidence which can be found in Figure 4f.

4. The authors raise an interesting hypothesis about IL and OL neurons: these neurons may be involved in contact with nematode prey and modulated by tyramine/octopamine to alter how animals respond to those incoming prey-related sensory cues. However, there really needs to be further experiments here. I'd suggest both (i) silencing these neuron classes and examining the behavioral impact; and (ii) performing genetic rescue of the receptors in these cells. This would provide the basic evidence necessary to support the authors' hypothesis.

This is similar to the concern raised by reviewer 2, and we agree that the potential divergent function of the sensory neurons such as the ILs acquiring modulation by octopamine and/or tyramine is fascinating. To explore this further we focused on the octopamine receptors which are the likely regulators of predatory aggression. These are expressed in the IL2s in *P. pacificus* but are absent in *C. elegans* revealing a potentially divergent function. Therefore, we firstly silenced these neurons as suggested by the reviewer. We expressed a histamine gated chloride channel (HisCl) specifically in these neurons which is not natively found in nematodes. This enables inducible inhibition of target neurons upon the exogenous addition of histamine. Alongside the recent publication by Mackie et al 2025 *elife*, this is the first successful use of this technique in *P. pacificus*. We identified a IL2 specific promoter (*Ppa-klp-6*) and upon silencing these neurons we were able to robustly phenocopy all the octopamine mutants (*tbh-1*, *ser-3* and *ser-6*) with a similarly drastic reduction in predatory aggression. In concurrent experiments (<https://www.biorxiv.org/content/10.1101/2025.03.24.644997v2>) we have now found that these IL2 neurons also express several sensory receptors involved in prey detection indicating a remarkable convergence of these processes into these sensory neurons.

5. I'm unclear how well conserved the amino acid sequences of the octopamine and tyramine receptors are. If they are not identical, are the authors sure that these receptors definitely respond to the same ligands? For example, some of these receptors (e.g. ser-3) were originally misnamed as presumed serotonin receptors based on homology, but subsequently shown to actually respond to different amines. The authors should comment on this and, if the sequences are divergent, provide evidence (genetic or biochemical) that the main receptor(s) of interest are indeed selectively activated by tyramine or octopamine.

We thank the reviewer for raising this issue. We agree that this is important information and therefore have now included the phylogenetic analysis and alignments for these genes comparing *C. elegans* and *P. pacificus* variants in the supplementary figures. We find that the neuromodulator biosynthesis enzymes and associated receptors are highly conserved with clear one to one homology across these species.

Minor comments:

1. Please add WT to Extended data fig. 5a.

Done

2. Please add WT to Fig. 4e

Done

3. There is a typo (missing character?) on line 342 that was confusing.

Done

4. I don't usually care about semantics, but I did find myself wondering whether aggression was really the correct term here. Why not consummatory behavior? Tbh-1 (at least in Fig. 3) has reduced biting, but also reduced dwelling. Dwelling is a non-aggressive consummatory behavior. Perhaps the authors could weigh alternatives and justify their terminology choice in the Results section.

We appreciate the reviewer's comment, which mirrors some of the points raised by reviewer 2. We have considered this point carefully, and based on recent literature (for example the PNAS paper <https://www.pnas.org/doi/10.1073/pnas.2422935122> which describes cannibalistic aggression in caterpillars) and our own findings, we are convinced that the observed states are indeed aggressive, rather than purely nutrient driven. While intraguild predation inherently serves both aggressive and nutritional goals, a hallmark of aggression is biting without feeding, as biting only serves to sever the cuticle, while feeding is the state during which substantial amounts of material are ingested. We therefore reanalyzed our data to extract two aspects: The number of biting events, and how frequently biting was followed by feeding. As we show in Fig. 2, biting is very specific to a prey context and motor patterns resembling biting are rarely if ever observed on bacteria alone. In contrast, on larval prey, we observe frequent bites, only some of which transition into feeding states. To further separate the nutritional and aggressive aspects, we have conducted new experiments supplying the predators with both prey and a bacterial food source. In this case, nutritional needs can be fully served by bacterial ingestion (as these animals can grow entirely on bacteria), and any biting would be likely aggressive. Indeed, we find that in this mixed condition, animals still bite larval prey at similar rates as when they only have prey present, but the transitions from biting to feeding states are reduced, indicating a substantial proportion of biting events represent purely aggressive acts.

We have clarified this important distinction in the main text (lines 168-185) and added these data as Extended Data Fig. 4d-f.

Referee #2 (Remarks to the Author):

The precise genetic, molecular, and neural mechanisms shaping species differences in behavioral variation are poorly understood, especially in the context of the evolution of novel behaviors. The evolution of intraguild predation behavior in *P. pacificus*, where individuals kill and prey on different species to both reduce competition and fulfill nutritional needs, provides an excellent model system. Here, the authors have developed genetic reagent that enables the application of a machine learning-based approach to define the various behavioral states associated with predatory behavior, providing an unbiased, high-throughput assay to examine the roles of noradrenergic pathways. Specifically, based on the mutant phenotypes of neuromodulators and their receptors, the authors provide evidence that the octopamine pathway promotes predatory biting, while the tyramine pathway acts antagonistically to induce non-predatory states. The authors further identified a few putative species differences, including an additional pair of octopaminergic neurons with neurite projecting to the mouth region associated with feeding. This neurite is likely absent in *C. elegans*. The authors also identified that this neuron potentially expresses an octopamine receptor, raising the possibility of a self-reinforcing motif that may contribute to a persistent predatory state. Finally, the authors identified possible species differences in the expression pattern of octopamine receptor neurons in *P. pacificus*, including neurons that project to head sensory neurons extending to the nose. Together, the authors suggest that noradrenergic circuits balance aggressive behavioral states in nematodes, and evolutionary adaptations in the octopamine/tyramine circuits facilitate the evolution of predatory behaviors, potentially by promoting a persistent aggressive state and state-dependent gating of sensory neurons. By addressing the neural mechanisms for a well-known behavioral innovation and the neuromodulatory mechanisms regulating such a complex behavior, the work is of potential interests to a broad audience.

We appreciate the reviewer's engagement with our work and their detailed evaluation. Below, we detail our responses, which we hope fully address their questions.

Major comments

The role of noradrenergic circuits in predatory behaviors in *P. pacificus* has been convincingly established in this study. Nevertheless, the main conclusion of this manuscript—that evolutionary adaptations in octopaminergic circuits have facilitated the emergence of aggressive behavioral states associated with complex predatory traits—has yet to be fully established. To convincingly support this conclusion, the following points need to be addressed: (1) clearly define the species differences; (2) examine the phenotypic outcomes of key species differences; and (3) demonstrate that these species differences represent predatory-specific adaptations. Additionally, caution is needed when interpreting whether the phenotypes observed in predatory behaviors (such as biting) can be directly considered aggressive states. Below, I will discuss each of these points in more detail.

We appreciate the structured comments, and summarize our revised support for our conclusions here, with additional details below.

- (1) To clearly define the species differences, we now directly compare the expression patterns between species using the same labelling modality, confirming the importance of the octopamine receptor expression divergence.
- (2) We have been able to pinpoint that this receptor divergence directly results in changes in predatory aggression when these cells are inhibited, establishing the crucial role of the IL2 neurons as point of divergence.
- (3) To determine that these adaptations represent lineage-specific changes in behavior, resulting in predatory aggression, we complemented our existing studies by investigating a basal member of the Diplogastrids. We find that similar to *P. pacificus* this basal species shares all predatory aggressive traits and indeed, these are regulated by octopamine.

First, the authors have identified several notable species differences. The most important is the gain of a pair of octopaminergic neurons. Specifically, the putative RIM is both tyraminerpic and octopaminergic in *P. pacificus*, while it is only tyraminerpic in *C. elegans*. This conclusion is based on a transgenic reporter assay, which has its limitations (e.g., genomic location effects). Given the importance of this finding for the manuscript's overall conclusion, further validation is required, such as a CRISPR knock-in of the reporter gene at its endogenous location. If HDR-directed CRISPR is not established in *P. pacificus*, can the authors consider alternative methods like in situ hybridization. Furthermore, the putative RIM neuron in *P. pacificus* exhibits a neurite extending anteriorly toward the mouth, a feature absent in *C. elegans*. Given the reported differences in both the noradrenergic identity and morphology of RIM between the species, the homology of RIM should be strengthened, such as by examining characteristic branching patterns of RIM, if any. A recently published preprint comparing the EM connectomes of the two species (Cook et al. 2024) noted the conservation of characteristic branching patterns in RIM but did not mention any species-specific pattern. Additionally, based on the results in this preprint, *P. pacificus*'s RIM did not seem to exhibit major *P. pacificus*-specific postsynaptic connections. While these EM data are not definitive, the potential inconsistency with the interpretation from the current study calls for caution of inferring homology. Strong evidence for homology is a necessary foundation for the authors' subsequent conclusions about rewiring and morphological divergence. Finally, the authors identified species differences in the expression patterns of octopamine receptors, with RIM potentially acting as a *P. pacificus*-specific site for a positive auto-receptor motif. The possibility that RIM functions as such a motif to promote a persistent predatory state is exciting but could benefit from simple validation (e.g., in situ hybridization to confirm that the putative RIM is also *ser-6* positive). Similarly, considering the different methods used for species comparison (reporter assays in *P. pacificus* and scRNA-seq in *C. elegans*), important candidate species differences (such as IL1 and IL2) should be further validated applying the same method.

We thank the reviewer for the detailed response and appreciate their insight. Importantly, establishing true neuronal identity between species with a consistent definition is still an open question. One view may be that developmental ontogeny is relevant, thus looking at transcriptional markers of identity or alternatively a functional view may consider instead the expression of channels and receptors to be of importance. However, similar to reviewer 1 point 3 we have now attempted to clarify the neuronal identity of the *tbh-1* and *tdc-1* expressing cells using multiple different methods. These include antibody staining, other transcriptional markers and HCR. We were able to successfully implement HCR in *P. pacificus* following a recent protocol (Ramadan and Hobert 2024). While transcriptional reporters may not fully contain all cis-regulatory elements, HCR similarly has disadvantages as it only shows a temporal snapshot of expression and is limited to showing transcripts in the soma. However, with our new HCR data we observe neuron expression similar to that observed in *C. elegans*, including 2 neuron pairs expressing *tdc-1*, and a single pair with *tbh-1*. We now present these data in Figure 4f.

To validate the essential species-specific adaptations, we have now further investigated the relevant receptors using the same techniques in both species. We have shown in transcriptional reporter lines that *ser-3* is present in the *P. pacificus* IL2 neurons, but absent in *C. elegans* IL2 neurons. These are now presented side by side in Figure 5g.

Second, despite identifying many potentially relevant species differences, the phenotypic outcomes of these differences were not directly examined. While I understand the technical challenges, functional validation is critical to make the stated inferences. Particularly, *C. elegans* and *P. pacificus* are highly divergent species (100-300 million years ago), and many changes may have occurred. For example, the EM connectome has identified widespread differences between the two species in cell number, projection patterns, cell body positions, and synaptic connectivity, but not all of these differences are expected to be relevant for the predatory behavior. Furthermore, not all changes are necessarily functional—developmental and circuit drift are common phenomena in evolution (for example, see the connectome paper for examples of developmental drift). At a minimum, the

authors should attempt to address the functional consequences of key species differences, such as using RIM-specific RNAi of *tbh-1* or *ser-6*. I understand that these experiments may or may not be readily possible, but without such support, the major conclusion of the paper lacks direct support.

We appreciate these insights and agree that it is crucially important to also understand the phenotypic implications of these changes which is also an issue raised by reviewer 1. Therefore, we have additionally investigated the acquisition of octopaminergic signaling in the IL2 neurons which are one of the most prominent differences between these species. These neurons express the octopamine receptor *ser-3* in *P. pacificus* but not in *C. elegans*.

To directly link the acquisition of the octopamine receptor in the IL2s with predatory aggression, we utilized a histamine gated chloride channel (HisCl) to silence these neurons specifically. (See also response to reviewer 1). This enables inducible inhibition of target neurons upon the exogenous addition of histamine. Alongside the recent publication by Mackie et al 2025 *elife*, this is the first successful use of this technique in *P. pacificus*. We identified a IL2 specific promoter in *P. pacificus* (*Ppa-klp-6*) and upon silencing these neurons we were able to robustly phenocopy the octopamine mutants (*tbh-1*, *ser-3* and *ser-6*) with a similarly drastic reduction in predatory aggression. In concurrent experiments (<https://www.biorxiv.org/content/10.1101/2025.03.24.644997v2>) we have now found that these IL2 neurons also express several sensory receptors involved in prey detection indicating a remarkable convergence of these processes into these sensory neurons.

Taken together, these data directly link the adaptations of the IL2 neurons with predatory aggression.

Third, do species differences observed readily represent *P. pacificus*/lineage-specific adaptations for its unique predatory behavior? I am afraid not. The notes species differences (e.g., the putative RIM being octopaminergic in *P. pacificus* but not in *C. elegans*) could be either a gain in *P. pacificus* or a loss in *C. elegans*. Phylogenetic support (e.g., outgroup species of *P. pacificus* without predatory feeding does not exhibit the additional pair of octopaminergic neurons) is needed to distinguish the two possibilities, especially given the long evolutionary divergence time.

We thank the reviewer and agree that with only a two-species comparison, it is difficult to infer the evolutionary history of these adaptations and their importance for the acquisition of the predatory aggression phenotypes. To overcome this, we have expanded our evolutionary analysis of octopamine and its role in predatory aggression to another member of the diplogastrid nematodes. This is the nematode family which *P. pacificus* belongs to and so far, nearly every species discovered is capable of predatory aggression. To assess the regulatory role of octopamine we made *tbh-1* mutants in *Allodiplogaster sudhausi* which is one of the basal representatives of this clade. This species recently underwent a whole genome duplication event and grows very slowly making CRISPR/Cas9 more complicated. However, due to its phylogenetic position and the fact one previous paper managed to generate CRISPR/Cas9 mutants we believed it was the best species to strengthen our argument. Here we showed *Asu-tbh-1* mutants (putative null mutant in both copies) also show a reduction in predatory aggression mirroring our findings in *P. pacificus*. This reinforces our initial conclusion and suggests an ancient and conserved role across the diplogastrid nematodes associating octopamine with predatory aggression. This is divergent from its known function outside of this nematode family and based on this additional evidence represents a lineage-specific adaptation.

Fourth, the authors switched from using the term 'predatory behavior' in the Results section to 'aggressive behavior' in the Discussion (Abstract as well) without providing a clear rationale for this interchange. This should be handled with great caution. First, although intraguild predation behavior involves aggression with the goal of killing the competitor, it also results in obtaining nutritional resources. Aggression has typically been distinguished from predation due to its nutritional goals (Quach and Chalasani et al., 2020). Predatory biting can be driven by both goals, and this distinction cannot be made from the current assay. Further, while a reasonable hypothesis, there is no direct

evidence supporting that the octopamine/tyramine circuits encode the aggressive state. To support the claim that octopaminergic neurons encode aggression (or that octopamine/tyramine neurons gate the aggressive state, as stated in the title of Figure 4), as the minimal, evidence such as neuron activation leading to persistent biting but not feeding is needed. Additionally, since octopamine injection did not increase biting (line 161, ED Fig. 6), interchanging the two terms is particularly concerning.

We appreciate the reviewer's comment, which mirrors some of the points raised by reviewer 1. We have considered this point carefully, and based on literature (for example the PNAS paper <https://www.pnas.org/doi/10.1073/pnas.2422935122> which describes cannibalistic aggression in caterpillars) and our own findings, we are convinced that the observed states are indeed aggressive, rather than purely nutrient driven. While intraguild predation inherently serves both aggressive and nutritional goals, a hallmark of aggression is biting without feeding, as biting only serves to sever the cuticle, while feeding states are when substantial amounts of material are ingested. We therefore reanalyzed our data to extract and differentiate between these two aspects including identifying the number of biting events, and how frequently biting was followed by feeding. As we show in Fig. 2, biting is very specific to a prey context and motor patterns resembling biting are rarely if ever observed on bacteria alone. In contrast, on larval prey, we observe frequent bites, only some of which transition into feeding states. To further separate the nutritional and aggressive aspects, we supplied the predators with both prey and a bacterial food source. In this case, nutritional needs can be fully served by bacterial ingestion (as these animals can grow entirely on bacteria), and any biting would be likely aggressive. Indeed, we find that in this mixed condition, animals still bite larval prey at similar rates as when they only have prey present, but the transitions from biting to feeding states are reduced, indicating a substantial proportion of biting events represent purely aggressive acts.

We have clarified this important distinction in the main text (lines 168-185) and Extended Data Fig. 4d-f.

Minor comments

Line 161: Can the authors speculate why octopamine injections didn't increase aggressive behavior?

We were also fascinated by this result which we interpret as the maximum predatory aggressive potential has been reached and the system is saturated.

We have now explored these findings further by including additional *tbh-1* and *tdc-1* mutants. The addition of octopamine into *tbh-1* and *tdc-1* null mutant background also rescues the killing phenotype reinforcing its role but does not increase this further, similar to the addition in the wildtype background. This is now included in Extended Data Fig. 6.

Line 162: The statement that "evolutionary divergence in the function of these neuromodulators" is potentially problematic, especially when "function" is not clearly defined. My opinion is that the term "evolutionary divergence" is typically not used in such context. For example, dopamine may promote different behaviors in a species-specific manner, but the function of dopamine as a rewarding signal remains the same, and the dopaminergic circuit also does not have to be functionally divergent. In the current study, the more direct interpretation is that "these neuromodulators regulate this novel behavior."

Done

Error bars are not defined in figure legends throughout the main and ED figures.

Done

Figure 3d: Dopamine-deficient mutants show decreased 'predatory search' and 'predatory feeding', but not 'predatory biting.' Based on this observation, the authors predicted that dopamine may be required for initiating feeding after a kill. This does not explain why 'predatory search' also decreases. It is also unclear why decreased 'predatory search' did not lead to decreased biting. Would this imply that the probability of biting following a search increases?

We agree with the reviewer that the dopamine phenotype is also of significant interest. Ongoing investigations are currently further analyzing the dopamine phenotype including all of the associated receptors and their function. As dopamine is also a major regulator of motoneurons, with motoneurons expressing both excitatory and inhibitory DA receptors, we believe that changes in motor patterns and possibly head swings may be at play here. As predatory search is characterized by head swings, we want to further dissect the role of dopamine in motor control versus reward signaling. However, these experiments represent an entirely new study and go beyond the scope of this paper.

Figures 4d and 3d: The phenotypes of *tbh-1* mutants appear quite different between the two figures? Would this difference be large enough to impact the robustness of the conclusions? For example, if the *tbh-1* data in figure 4 were replaced with those from figure 3, would the conclusions of all the comparisons remain the same?

We apologize for the confusion regarding this point. As we have stated in the response to reviewer 1, we have clarified the statistics used in these two figures. In the first case, we compare vs wt, and in Fig. 4 we compare vs *tbh-1*. Therefore, the phenotype is consistent in both figures using independent data sets and there is no impact on the outcome or our conclusions.

Figure 4e: The role of *ppa-1gc-55* is the strongest among the four receptors; however, other tyramine receptors may also play a role. Not fully restoring the phenotype to the wild-type level does not necessarily mean that they don't play a role. The authors stated that double mutants of *tbh-1* and individual tyramine receptors were tested to see if any tyramine receptor could rescue the *tbh-1* phenotype in reducing aggressive biting. If so, why were all the comparisons made to *tdc-1* (to test for full rescue) instead of *tbh-1* (to test for any rescue)?

While we observed a rescue in the *lgc-55* mutants, our analysis of a triple mutant of the remaining tyramine receptors *ser-2*; *tyra-2*; *tyra-3* failed to rescue the phenotype with no detectable increase in predatory aggression (Extended Data Figure 8e-g). We believe that it is most likely that *lgc-55* is the relevant receptor involved in this pathway. To further clarify this, we have included *tbh-1* mutants as a control alongside the *tbh-1*; *ser-2*; *tyra-2*; *tyra-3* mutants. This data in Extended Data figure 9e - g.

Line 225: The authors suggest that the auto-receptor motif may contribute to persistent and robust predatory episodes. If so, would the measurement of "state duration" (in addition to "state time") also be directly relevant? However, for the two feeding conditions in wild-type animals, the "state duration" remains largely unaltered (ED Fig. 4).

The text has been changed according to our new HCR data.

Are the functions of the sensory neurons that express octopamine receptors known? If relevant, this may provide support for the hypothesis of state-dependent gating of sensory neurons in regulating aggressive behavior. If not, the tone for this interpretation shall be greatly weakened at least.

Similar to our response to reviewer 1, we have been concurrently investigating the role of the *ser-3* expressing IL2 neurons (<https://www.biorxiv.org/content/10.1101/2025.03.24.644997v2>). These

neurons express a number of mechanosensory and chemosensory receptors required for prey detection. Together with our result of octopaminergic gating these cells, this represents a remarkable convergence of these processes into these sensory neurons.

To strengthen our conclusions regarding the vital role of the IL2 neurons, we performed additional silencing experiments in these cells. We find that silencing the IL2s substantially reduces predatory aggression and phenocopies the *tbh-1* mutant.

Line 399: Method: "predictions with a probability of less than 50% are set to 'None'." Can the authors provide a rationale for this threshold? How is the probability threshold for calling a prediction decided? Is it based on balancing precision and recall? I could not find relevant information. Please include it if it is indeed missing.

We found that the majority of our predictions (>99%) falls above 90% thresholds, indicating that the model can easily discriminate the behavioral states. 50% is a typical threshold used in similar models which balances the false-positive rates with a reasonable level of unlabeled frames. In our case, we only have 1.2% of unlabeled frames using this specific threshold. We have added this information to the methods section under *Behavioral state classification*.

Figures 4g and 4h could benefit from improved readability by (1) labeling neurons of central interest in this study (RIM, IL1, IL2) in *P. pacificus* and (2) labeling where the head is.

We have added labels to new figures 4f and 5f as requested.

Line 120: "Thus, we observe greatly expanded state complexity in *P. pacificus*, including the presence of aggressive episodes, which have not been previously reported in other nematodes, including *C. elegans*." I don't quite understand this sentence since *C. elegans* does not exhibit these behaviors. For me, it simply means that "the pipeline successfully captures the various states associated with predatory behaviors in *P. pacificus*."

Changed to "we observe a greatly expanded state complexity in *P. pacificus*, going beyond the canonical foraging switch between roaming and dwelling found in *C. elegans*"

Ground truth for machine learning: How do the authors assign states, and what is the source of the ground truth? For example, is the assignment of 'predatory biting' blind to information on the presence/absence of *C. elegans* larvae? In other words, would any clips from the bacterial lawn be defined as 'predatory biting' and 'predatory feeding'? If so, what percentage of these clips come from the *C. elegans* larvae condition compared to other conditions?

See our reply to reviewer 1, we have now added further validation to the specificity of the predatory state predictions which are consistent and robust across distinct data sets (see Extended data Fig 2d-f). We have also added an additional figure relevant to this new analysis (see figure 3).

Could the authors provide a panel comparing RIM cell body location and morphology between species side by side and noting the *P. pacificus*-specific phalangeal-projecting neurite?

Figure changed according to new HCR data.

Figure 1f: I guess this panel represents the average of all states? If so, please note it in the figure legend. Also remove the redundant data from Extended Data Figure 3.

We have added to the figure legend and removed the redundant data from the extended data.

Line 288: "two pairs of neurons expressing both neurotransmitters with novel features"— RIM is the

only one with novel features?

Text changed according to new HCR data.

Referee #3 (Remarks to the Author):

This study was a total pleasure to read. They ask penetrating questions, use powerful tools with precision and thoroughness, expand our understanding of noradrenergic pathway from one nematode species to another, and discover interlocking patterns involving evolutionary conservation and divergence at all levels from signaling molecules to receptors to cells to behaviors.

Pristionchus pacificus has a fraction of the tools that are available in *C. elegans*, and yet they manage (with CRISPR) to effectively dissect its noradrenergic circuits. I marvel at the intelligence of their new behavioral and segmentation assays. I marvel at the clarity and relentless logic of their prose.

The authors study the neural and molecular basis of *Pristionchus* predation. They express RFP in the pharyngeal muscles (unlike in *C. elegans*, the fluorescence is only visible in the corpus) and record animals in varying environments and under various genetic perturbations. The paper leverages a previously built pipeline called PharaGlow (Bonnard et al. 2022), which allows the authors to extract features from the recordings of the fluorescently labeled pharyngeal muscles. Via unsupervised clustering (UMAP followed by hierarchical clustering), the authors identify six distinct behavioral states, corresponding to roaming, dwelling, searching, predatory feeding, predatory biting and predatory searching. An expert annotator then creates ground truth for recordings and a classifier is trained to label recordings into one or none of these six states. The authors use plots such as the probability density of pumping rates and velocities, the transition probabilities between states and the percentage time an animal spends in a certain state to study how environment and neuromodulators affect predatory behavior.

The authors observe that *Pristionchus* predation related behaviors almost exclusively occur on larvae. The authors then ask if neuromodulators play an interesting role in this transition, and screen four mutants based on orthologues in: *\emph{tph-1}* (serotonin), *\emph{cat-2}* (dopamine), *\emph{tbh-1}* (octopamine) and *\emph{tdc-1}* (tyramine, and by proxy octopamine). Consistent with previous literature, they find similar implications of serotonin in predatory behavior.

The authors also find that *\emph{cat-2}* mutants have a reduction in predatory feeding and predatory search, but not in biting behavior, implying that dopamine could be a food reward signal the animal receives after a successful kill. Interestingly, the authors also show the role of octopamine in the induction of a prolonged predatory state, and tyramine in the induction of a docile state. They identify neurons RIM and RIC as both tyraminergetic and octopaminergic, and observe an extra neurite in RIM that makes it distinct from the *C. elegans* RIM.

The authors then study receptors of octopamine and tyramine by making mutants of orthologues of known *C. elegans* receptors.

They observe that cells expressing *\emph{ser-3}* and *\emph{ser-6}* octopamine receptors are important in predation via the octopamine signaling pathway. Cells expressing *\emph{lgc-55}*, a tyramine receptor, are important in the docile state induced by the tyramine signaling pathway. Using reporter lines, they observe that these receptors are expressed in neck muscles and a few head neurons. Interestingly, octopamine receptors can be found in IL1 (mechanosensory), IL2 and RIM (expression in RIM implies some feedback). Tyramine receptors seem to be expressed in a distinct set of head neurons, one of which seems to be OL.

All I really had to do in this Review is document their beautifully done work. In the end, my only real question was how they might prove that the bistable state-switch (octopamine release stimulating

further octopamine release), actually works this way. It suggests a truly elegant cell-autonomous mechanism for sustaining an octopamine-dependent behavioral state. The hypothesis is reasonable, and I couldn't think of a reasonable experiment that would cleanly test the hypothesis. The bistable state-switch model is adequately presented as food for thought.

In closing, this is a seriously well-done study and well-written paper. Congratulations!

We thank the reviewer for their very positive response to our manuscript!

[REDACTED]

We would wish to sincerely thank the reviewers for their constructive and insightful feedback on our manuscript. We are pleased that Reviewers 2 and 3 endorse publication, and we have made a concerted effort to address the remaining points raised by Reviewer 1. We have now carefully considered and addressed their issues to the fullest extent possible given the current state-of-the-art methods and have revised the manuscript accordingly. We believe these revisions have resolved most of the outstanding issues and have further strengthened the manuscript. We are grateful for the opportunity to resubmit our improved version to *Nature*.

Kind regards,
James Lightfoot and Monika Scholz
on behalf of all authors

1. Which neurons make tyramine and octopamine in *P. pacificus* to regulate behavioral states? In the initial submission, there were claims that RIM produced both tyramine and octopamine and had an anterior projection. These very surprising findings were removed from the study. I'm guessing that the authors no longer believe those results, which is fine. What is left in the revised manuscript is HCR labeling of *tdc-1* and *tbh-1*. While the number of labeled neurons now matches the *C. elegans* results, the labeling of neuronal identity from these images is at best a guess (nobody could confidently label the identities of these cells from those pictures). In addition, the authors did not provide requested genetic rescues of *tdc-1* or *tbh-1* in RIM or RIC. Unfortunately, this leaves a pretty central part of the model unresolved. Is RIC the functional site of octopamine production relevant to these behaviors? Is RIM the functional site of tyramine production? The final model cartoon in the paper (Fig.6i) depicts it as if this was shown to be the case. I hate to add extra work, as I appreciate that the authors have already done a lot, but I do think the bare minimum to resolve this is important. There are abundant reporter gene studies for other genes and transgenic lines (for HisCl expression, etc) elsewhere in the study. I'd suggest the author confirm that RIM and RIC are the relevant neurons that produce these neuromodulators (via *tdc-1* and *tbh-1* reporter genes or HCR labeling overlapping with genetic markers for those 2 cells). In addition, they should silence RIM and RIC each on their own (or perform cell-specific rescues of *tdc-1* and *tbh-1*, respectively) and examine the consequences for behavioral states.

We thank the reviewer for their suggestions and apologise for the lack of clarity regarding RIM and RIC identity. We have now included an additional *tdc-1* transcriptional reporter line (Fig. 4f). This shows two pairs of neurons expressing *tdc-1* that match the morphology and soma location of the neurons described in the recent *P. pacificus* connectome paper (Cook et al, 2025, *Science*). In addition, a recent pre-print (<https://www.biorxiv.org/content/10.1101/2025.10.16.682888v1>) has just been published online led by the labs of Oliver Hobert (Columbia, NY) and Ray Hong (Northridge, Ca) that also identify these neurons as RIC and RIM and validated them with *cat-1* transgenes, HCR and DAPI imaging. All of these are consistent with the same cells in *C. elegans*. We have added these additional experiments, comments and references to the text, that taken together strongly support these neuronal identities as RIC and RIM.

As *tbh-1* and *tdc-1* encode the rate-limiting enzymes for octopamine and tyramine synthesis, respectively, and we confirmed their expression using HCR and transcriptional reporters, our results strongly suggest that RIC is the principal source of octopamine, while both RIC and RIM contribute to tyramine production. Given that no additional neuronal sources of

octopamine have been identified, this model (Fig. 6) represents the most parsimonious interpretation of the available data. We thank the reviewer for prompting clarification on this point and have revised the text to more explicitly communicate that these cells are the likely sources of these amines in nematodes, where biogenic amine synthesis is typically confined to a very small number of neurons.

Definitive functional validation of these sites, however, remains technically challenging due to the current lack of neuron-specific promoters in *P. pacificus*. This limits our ability to selectively manipulate RIC and RIM. To address this, we have included a brief summary of *C. elegans* RIC and RIM markers and the corresponding complexities in *P. pacificus*, highlighting the experimental constraints:

RIC

RIC specific promoters are difficult as even in *C. elegans* *tbh-1* reporters are sometimes reported to have weak and ectopic expression in other cells. The cleanest RIC marker used in *C. elegans* is achieved using the 3rd intron of *ser-2*. This results in strong RIC expression but also occasional weak expression in other head neurons as well. This intron has diverged between *C. elegans* and *P. pacificus* showing no homology and has no predicted enhancer activity according to recent *P. pacificus* chromatin state studies (Werner et al, 2018).

gcy-13 – expressed in RIC and several other neurons. No one to one homology as this gene has undergone a large expansion in *P. pacificus* resulting in 10 paralogues.

ceh-17 - RIC and several other neurons. Has no obvious homologue in *P. pacificus*.

flp-18 – RIC and several head sensory neurons including AIY.

RIM

RIM markers used in *Cel* include the following:

The *F23H12.7* promoter is currently the best option in *C. elegans* however it has no one to one homologue in *P. pacificus* making it unusable for our studies.

More recently, an additional RIM specific reporter *cex-1* has also been described in *C. elegans* (Li et al., 2023, *Mol. Neurosci*). This does have one to one homology with *P. pacificus*. We have attempted to use this however its expression has diverged and a *Ppa-cex-1::RFP* line is instead expressed in a different head sensory neuron in *P. pacificus* that is anterior to our RIC and RIM marker and is likely an amphid cell due to the presence of the anterior projecting neurites (see below).

nmr-1 and *nmr-2* are used in other *C. elegans* studies; however, these are also expressed in several additional head neurons making this unsuitable.

Therefore, to confirm RIM and RIC are the relevant neurons we have instead silenced both cells simultaneously by using the *tdc-1* promoter to drive HisCl expression in these cells. This should recapitulate the *tdc-1* phenotype that is wildtype levels of killing. Thus, we have conducted these experiments in the *tbh-1* mutant background in order to detect this change. Indeed, we observe a partial rescue of the *tbh-1* phenotype that moderately recapitulates the *tdc-1* mutant. The partial rescue is likely due to the relatively low expression we observe using the *tdc-1* promoter (see fig 4f,g). These data indicate that these neurons are indeed the relevant functional site of tyramine production for this behaviour.

This is now included as a new Extended Data Fig. 7.

2. The IL2::HisCl experiment is nice. But it is still not clear that octopamine receptor expression in IL2s has any functional importance. Given that the authors have a good IL2-specific promoter, could the authors perform a genetic rescue of key octopamine receptors and see if this rescues behavior? I had also suggested this experiment in my previous review.

We apologise for the omission of this experiment. We attempted the *ser-3* rescue using the *klp-6* IL2-specific promoter, which we had successfully utilized for HisCl-mediated genetic silencing of these neurons, to drive *ser-3* expression in these IL2 cells. In total, we injected 6,132 animals and recovered only four transgenic individuals that arrested during early development and a single adult that failed to produce transgenic offspring. Thus, we suspect that excessive *ser-3* expression driven by the *klp-6* promoter in the IL2 cells is detrimental to *P. pacificus* development and health. Curiously, during our previous experiments to silence the IL2 neurons, we also attempted a Caspase mediated genetic ablation of these cells using both *klp-6::Caspase3* and *klp-6::ICE*. In both cases, we were unable to recover transgenic lines as these animals also arrested in the larval stages. This may indicate these neurons are much more important for *P. pacificus* development than *C. elegans* which is something we will investigate further in the future. In *C. elegans*, such issues could potentially be overcome using methods for single-copy insertion such as MosSCI or MiniMos to reduce the copy number and better control expression levels. Both the Lightfoot and Scholz laboratories are actively developing these approaches for *P. pacificus*, but they are not yet operational. Thus, these results suggest that, due to current methodological limitations in the *P. pacificus* molecular toolkit, this experiment is technically not possible at present.

Importantly, while this experiment would certainly further substantiate the functional relevance of *ser-3* receptors in IL2 neurons, we do not believe its absence substantially weakens our findings or conclusions. In concert, we have also improved the summary figure in Fig. 6I to clearly indicate that other neurons expressing *ser-3* may also contribute, but couldn't be conclusively identified.

Again, I hate to ask for more work. But the model cartoon in Fig. 6i that is meant to summarize the paper indicates that RIM/RIC release tyramine/octopamine to signal to IL2/OL neurons via specific receptors SER-3/SER-6/LGC-55. Currently, this simply isn't supported by data and I don't see how I can suggest publication with that being the case. I am happy to recommend publication in Nature once these last questions are resolved.

We agree with the review that some aspects of the model are not yet supported. We have now tempered our model, specifically in regards to the tyramine ser-6, lgc-55 receptors and IL1 and OL predictions which at this stage are more speculative as they have not yet been validated with neuronal silencing experiments. We are excited to clarify this aspect further in future studies that will instead have a stronger focus on the docile states, their regulation and importance. We hope that with these changes the model accurately reflects our findings of the pathway and its implications for aggressive predatory behaviour.

[REDACTED]

We sincerely thank the final reviewer for the positive assessment and recommendation for publication and greatly appreciate their time and effort during this process. Please find our response to the comments from reviewer 1 below.

We are grateful for the opportunity to resubmit our improved version and look forward to seeing the work come to completion in *Nature*.

Kind regards,
James Lightfoot and Monika Scholz
on behalf of all authors

Referee #1 (Remarks to the Author):

I am happy to now recommend publication. I appreciate the *tdc-1::HisCl* experiment and best efforts with *ser-3*. It would be worthwhile to describe detailed methods for transgenesis in the Methods section of this study, as it sounds like the authors are leading experts and some challenges remain. Detailed methods could help others further develop this exciting model system.

Congratulations on a landmark study.

We are pleased that the additional *tdc-1::HisCl* experiment and our efforts with *ser-3* were appreciated.

As suggested, we have added an additional sentence to the Methods section to include the number of worms required to successfully generate these complex fluorescent strains as a guide for the growing *Pristionchus* community. Given the technical challenges of transgenesis in this system, we agree that these details will be valuable for others. Additionally, we are continuing to improve these methods in our own respective groups and hope to report substantial methodological improvements in the future.

We thank the reviewer once again for their supportive comments and are grateful for the recognition of our work as a landmark study.